# Prickle isoforms determine handedness of helical morphogenesis

**Bomsoo Cho, Song Song, Jeffrey D Axelrod\***

Department of Pathology, Stanford University School of Medicine, Stanford, United States

**Abstract** Subcellular asymmetry directed by the planar cell polarity (PCP) signaling pathway orients numerous morphogenetic events in both invertebrates and vertebrates. Here, we describe a morphogenetic movement in which the intertwined socket and shaft cells of the *Drosophila* anterior wing margin mechanosensory bristles undergo PCP-directed apical rotation, inducing twisting that results in a helical structure of defined chirality. We show that the Frizzled/Vang PCP signaling module coordinates polarity among and between bristles and surrounding cells to direct this rotation. Furthermore, we show that dynamic interplay between two isoforms of the Prickle protein determines right- or left-handed bristle morphogenesis. We provide evidence that, Frizzled/Vang signaling couples to the Fat/Dachsous PCP directional signal in opposite directions depending on whether $Pk^{pk}$ or $Pk^{sple}$ predominates. Dynamic interplay between Pk isoforms is likely to be an important determinant of PCP outcomes in diverse contexts. Similar mechanisms may orient other lateralizing morphogenetic processes.

## Introduction

PCP signaling controls the polarization of cells within the plane of an epithelium, orienting asymmetric cellular structures, cell divisions and cell migration. In flies, PCP signaling controls the orientation of trichomes (hairs) on the adult cuticle, orientation of ommatidia in the eye, and orientation of cell divisions, though the full range of phenotypic outputs has not been explored. While much focus has been placed on mechanistic studies in flies, medically important developmental defects and physiological processes in vertebrates are also under control of PCP signaling, motivating mechanistic studies in flies that might inform similar studies in vertebrates. Defects in the core PCP mechanism result in open neural tube defects, conotruncal heart defects, deafness, situs inversus and heterotaxy (reviewed in *Butler and Wallingford, 2017*; *Henderson et al., 2018*; *Blum and Ott, 2018*). PCP is also believed to participate in both early and late stages of cancer progression and in wound healing. PCP polarizes skin and hair, the ependyma and renal tubules. Paralogs of the PCP component Prickle are mutated in an epilepsy-ataxia syndrome (*Tao et al., 2011*; *Mei et al., 2013*; *Bassuk et al., 2008*; *Ehaideb et al., 2014*; *Paemka et al., 2015*). Mutations in 'global' PCP components have been associated with a human disorder of neuronal migration and proliferation (*Zakaria et al., 2014*) and in developmental renal disorders (*Zhang et al., 2019*).

Work in *Drosophila* indicates that at least two molecular modules contribute to PCP signaling. The core module acts both to amplify molecular asymmetry, and to coordinate polarization between neighboring cells, producing a local alignment of polarity. Proteins in the core module, including the serpentine protein Frizzled (Fz), the seven-pass atypical cadherin Flamingo (Fmi; a.k.a. Starry night), the 4-pass protein Van Gogh (Vang; a.k.a. Strabismus), and the cytosolic/peripheral membrane proteins Dishevelled (Dsh), Diego (Dgo), and the PET/Lim domain protein Prickle (Pk) adopt asymmetric subcellular localizations that predict the morphological polarity pattern such as hairs in the fly wing (reviewed in *Zallen, 2007*; *Butler and Wallingford, 2017*). These proteins communicate at cell boundaries, recruiting one group to the distal side of cells, and the other to the proximal side,

**\*For correspondence:**
jaxelrod@stanford.edu

**Competing interests:** The authors declare that no competing interests exist.

**eLife digest** Our right and left hands are mirror images of each other and cannot be precisely superimposed. This property, known as chirality, is vital for many tissues and organs to form correctly in humans and other animals. For example, fruit flies have hair-like sensory organs on the edges of their wings known as bristles. One of the cells in each bristle forms a shaft that generally tilts away from the main body of the fly and is anchored in place by another cell known as the socket.

A signaling pathway known as PCP signaling controls the directions in which many chiral tissues and organs in animals form. The pathway contains two signaling modules: the global module collects "directional" information about the orientation of the body and sends it to the core module, which interprets this information to control how the tissue or organ grows.

Fruit flies have two different versions of one of the core module components – known as Prickle and Spiny legs – that are thought to alter the direction the core module responds to the information it receives. Mutant flies known as $pk^{pk}$ mutants are unable to make Prickle and their wing bristles tilt in the opposite direction compared to those in normal flies, but it was not clear exactly why this happens.

To address this question, Cho et al. studied PCP signaling in the wings of normal and $pk^{pk}$ mutant flies. The experiments showed that Prickle directed the bristles on the right wing of a normal fly to grow in left-handed corkscrew-like patterns in which the emerging shaft and socket of each bristle twisted around each other. As a result, the bristles tilted away from the bodies of the flies. In the $pk^{pk}$ mutants, however, Spiny legs substituted for Prickle, causing the equivalent bristles to grow in a right-handed corkscrew pattern and tilt towards the body.

The findings of Cho et al. show that PCP signaling controls the direction fly bristles grow by selectively using Prickle and Spiny legs. In the future, this work may also aid efforts to develop effective screening and treatments for birth defects that result from the failure of chiral tissues and organs to form properly.

through the function of an incompletely understood feedback mechanism, thereby aligning the polarity of adjacent cells. A global module serves to provide directional information to the core module by converting tissue level expression gradients to asymmetric subcellular Fat (Ft) - Dachsous (Ds) heterodimer localization (reviewed in *Matis and Axelrod, 2013*; *Butler and Wallingford, 2017*; *Zallen, 2007*). The atypical cadherins Ft and Ds form heterodimers which may orient in either of two directions at cell-cell junctions. The Golgi resident protein Four-jointed (Fj) acts on both Ft and Ds as an ectokinase to make Ft a stronger ligand, and Ds a weaker ligand, for the other. Graded Fj and Ds expression therefore result in the conversion of transcriptional gradients to subcellular gradients, producing a larger fraction of Ft-Ds heterodimers in one orientation relative to the other. Other less well defined sources of global directional information appear to act in partially overlapping, tissue dependent ways (*Wu et al., 2013*; *Sharp and Axelrod, 2016*).

Various *Drosophila* tissues depend primarily on either of two isoforms of Pk, Prickle$^{prickle}$ (Pk$^{pk}$) and Prickle$^{spiny-legs}$ (Pk$^{sple}$) (*Gubb et al., 1999*). These isoforms have been proposed to determine the direction in which core PCP signaling responds to directional information provided by the Ft/Ds/Fj system (*Olofsson and Axelrod, 2014*; *Hogan et al., 2011*; *Ambegaonkar and Irvine, 2015*; *Ayukawa et al., 2014*). Pk$^{sple}$ binds directly to Ds, orienting Pk$^{sple}$-dependent core signaling with respect to the Ds and Fj gradients (*Ayukawa et al., 2014*; *Ambegaonkar and Irvine, 2015*), while Pk$^{pk}$-dependent core signaling has been proposed to couple less directly through a mechanism in which the Ft/Ds/Fj module directs the polarity of an apical microtubule cytoskeleton on which vesicles containing core proteins Fz, Dsh and Fmi undergo directionally biased trafficking (*Matis et al., 2014*; *Olofsson and Axelrod, 2014*; *Shimada et al., 2006*; *Harumoto et al., 2010*). Others, however, have argued that Pk$^{pk}$-dependent core signaling is instead uncoupled from the Ft/Ds/Fj signal (*Merkel et al., 2014*; *Ambegaonkar and Irvine, 2015*; *Casal et al., 2006*; *Lawrence et al., 2007*; *Brittle et al., 2012*).

The most intensively studied morphogenetic responses to PCP signaling in *Drosophila* occur in epithelia such as wing and abdomen, in which cellular projections called trichomes (hairs) grow in a

polarized fashion from the apical surface, and in ommatidia of the eye, in which photoreceptor clusters achieve chirality via directional cell fate signaling. Chaete (bristles), which serve as sensory organs, are also polarized by PCP signaling (*Schweisguth, 2015*; *Gubb and García-Bellido, 1982*). Bristles comprise the 4–5 progeny of a sensory mother cell (SMC), one of which, the shaft, extends a process above the epithelium such that it tilts in a defined direction with respect to the tissue. In mechanosensory microchaete on the notum, one daughter of the SMC divides to produce the shaft and a socket cell that surrounds the shaft where it emerges from the epithelial surface; the other SOP daughter divides to produce a glial cell, a sheath cell and a neuron. Studies of microchaete polarity have shown that the initial division of the SMC is polarized by PCP in the epithelium from which it derives, such that the two daughters are born in defined positions with respect to each other (*Gho and Schweisguth, 1998*; *Bellaïche et al., 2001*). However, subsequent events that ultimately determine the direction of shaft polarity have not been described.

We chose to study bristle polarity on the anterior margin of the wing (AWM). A row of stout mechanosensory bristles and a row of curved chemosensory bristles are on the dorsal surface, and a mixed row of mechano and chemosensory bristles is on the ventral side (*Figure 1A*, *Figure 1—figure supplement 1A*). All of these bristles tilt toward the distal end of the wing in wildtype. In $pk^{pk}$ mutant wings, and in wings overexpressing Pk$^{sple}$, a large fraction of the AWM bristles point proximally rather than distally; the $pk^{pk}$ phenotype is suppressed by mutation of *dsh*, implicating the core PCP signaling mechanism in this process (*Gubb et al., 1999*). However, the morphogenetic process resulting in polarity and the genetics of its apparent control by PCP signaling have not been explored. Here, focusing on the dorsal mechanosensory bristles, we report our analysis of the underlying morphogenesis leading to AWM bristle polarization, and show that polarization results from a corkscrew-like helical morphogenetic process involving the shaft and socket cells. Furthermore, our results reveal how interplay between Pk$^{pk}$ and Pk$^{sple}$ control the handedness of the helical growth, and how the Ft/Ds/Fj system directs it in opposite orientations depending on whether the core PCP mechanism operates in a Pk$^{pk}$- or Pk$^{sple}$-dependent mode.

## Results

### Proximo-distal relationship of wing mechanosensory bristle shaft-socket cell pairs is reversed in $pk^{pk}$ mutants

To begin to characterize the determinants of AWM mechanosensory bristle polarity, we labeled the externally exposed socket and shaft cells in wildtype (wt) and $pk^{pk}$ mutants with anti-Su(H) (Suppressor of hairless) antibody and phalloidin, respectively. In wildtype control wings at 36 hr apf, the apical ends of socket cells are tilted toward the distal end of the wing and are interspersed with the actin bundles of the shafts (*Figure 1C,C'*, *Figure 1—figure supplement 1B–B''*). The shafts are interspersed between socket cells, and appear to ascend along the proximal side of the adjacent socket cell and pass through an opening at its apical surface. Consistent with polarity patterns of adult bristles, actin bundles of $pk^{pk}$ mutant shafts near the distal end of the wing show a reversed, proximal, tilt, whereas shafts in the proximal and the very most distal regions show the normal distal tilt (*Figure 2A,A'*, *Figure 1—figure supplement 1C–C''*). Between these regions, shafts show a smooth transition between proximal and distal tilt, with some shafts pointing straight up (neutral tilt). The socket cells tilt at angles that correlate with the polarity of neighboring actin bundles in $pk^{pk}$ mutant wings: P-D tilt at the proximal region and D-P tilt at the distal region, with smooth transition between those regions (*Figure 1—figure supplement 1C–C''*). Furthermore, shaft actin bundles with reversed polarity appear to be positioned on the distal side of the socket cell through which they pass, opposite to their relationship in wildtype and to their relationship in the proximal region in $pk^{pk}$ mutant wings where their tilt shows the normal, distal, direction (*Figure 1—figure supplement 1B–B''*, compare with *Figure 1—figure supplement 1C–C''*).

Previous studies of AWM bristle ultrastructure have been insufficiently detailed to appreciate the determinants of polarity (*Hartenstein and Posakony, 1989*; *Palka et al., 1979*). To better understand the structures of shaft-socket pairs, and to unambiguously determine the relationship between sibling shaft-socket cell pairs, random individual bristle lineages were labeled by clonal expression of RFP, filling the cell bodies of labeled shaft and socket cells. Simultaneous staining of the socket cells (identified by Su(H) expression) allows one to identify the sibling shaft-socket cell pairs. 3D confocal

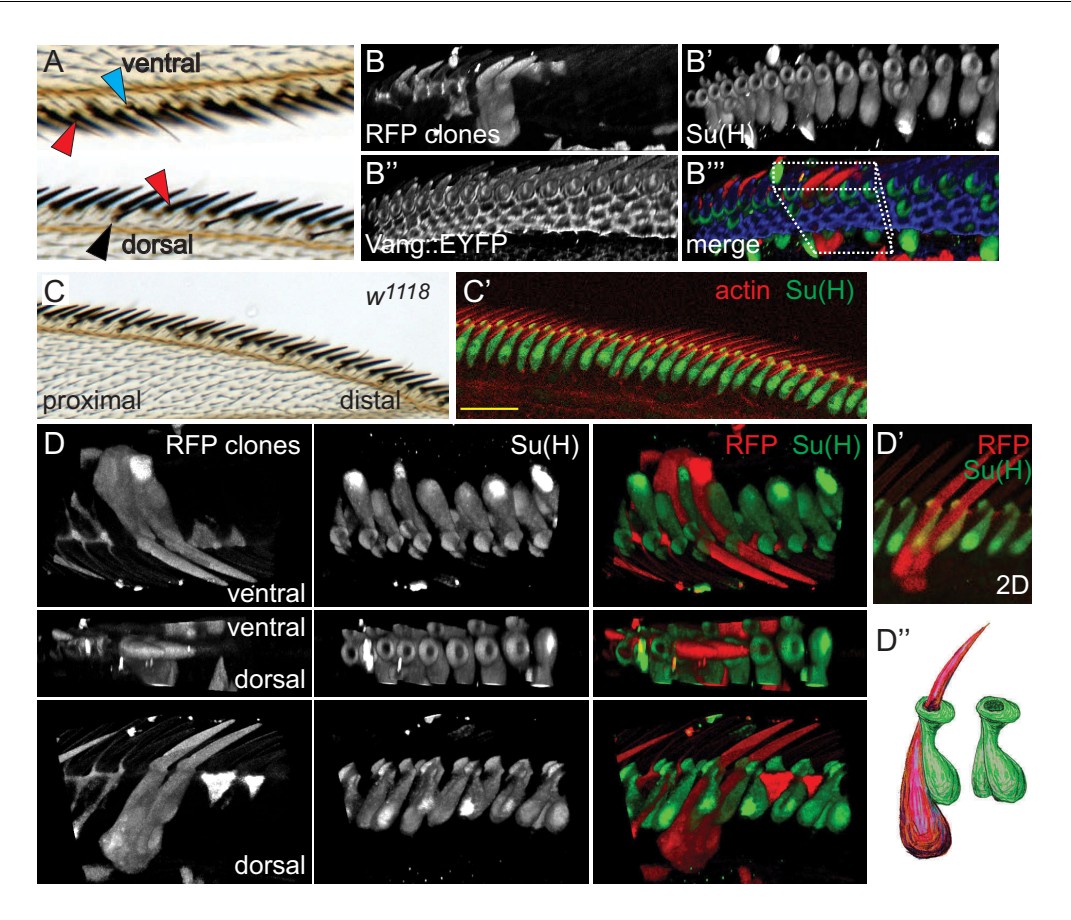

**Figure 1.** Morphology of wildtype dorsal mechanosensory bristles. (**A**) Dorsal and ventral views showing adult dorsal mechanosensory bristles (red arrowheads), ventral chemo- and mechanosensory bristles (blue arrowhead) and dorsal chemosensory bristles (black arrowhead). Bristles are separated by the exoskeleton secreted by cells displaying trichomes (hairs), similar to those in the majority of the wing blade. See also *Figure 1—figure supplement 1A* for a schematic view. (**B**) 3D reconstruction of a section of a 36 hr control ($w^{1118}$) AWM containing two clones expressing cytoplasmic RFP, each labeling two adjacent dorsal mechanosensory bristles. Several clones labeling hair cells are also present in this sample. Costaining with Su(H) marks all socket cells, and Vang::EYFP is present at apical cell junctions. The boxed region is displayed from several angles in panel D). (**C–C'**) Wildtype adult wing and equivalent region of a 36 hr pupal wing stained for Su(H) and actin. (**D–D'**) 3D views from different angles of the RFP clone(s) shown in panel B. (**D''**) Cartoon interpretation of the sibling shaft-socket pairs from **D'**. All images throughout are of right wings and are displayed proximal to the left and distal to the right. Scale bars: 20 μm.

The online version of this article includes the following video and figure supplement(s) for figure 1:

**Figure supplement 1.** Schematic of the AWM bristles, and expansive views of adult and pupal control ($w^{1118}$) and $pk^{pk30}$ bristles.

**Figure 1—video 1.** Morphology of 36 hr wildtype dorsal mechanosensory bristles displayed in 3D A reconstruction of confocal stacks displaying AWMs stained with Su(H) (green) to mark socket cells and RFP marking two shaft-socket cell pairs.

https://elifesciences.org/articles/51456#fig1video1

reconstructions facilitate examination from a variety of viewpoints and enable the positions and shapes of cell bodies, nuclei, shaft, and apical opening of the socket cells to be visualized (*Figure 1B–B'''*, *Figure 1—video 1*). These views allow us to see that the wildtype nucleus and main portion of the shaft cell is dorsal and extends slightly posterior to that of the sibling socket cell. The base of the shaft rises from the cell body, wraps clockwise along its socket sibling (as viewed from the apical side in a right wing), and then rises through a groove in the socket that extends apically along the proximal side of the socket, finally emerging through the donut shaped apical surface of the socket cell (*Figure 1D–D''*, *Figure 1—video 1*). In contrast, in the distal bristles of $pk^{pk}$ mutant wings where bristles are reversed, the shaft is positioned within an oppositely oriented groove on the distal side of the sibling socket cell. The shaft nucleus is dorsal and posterior to the socket

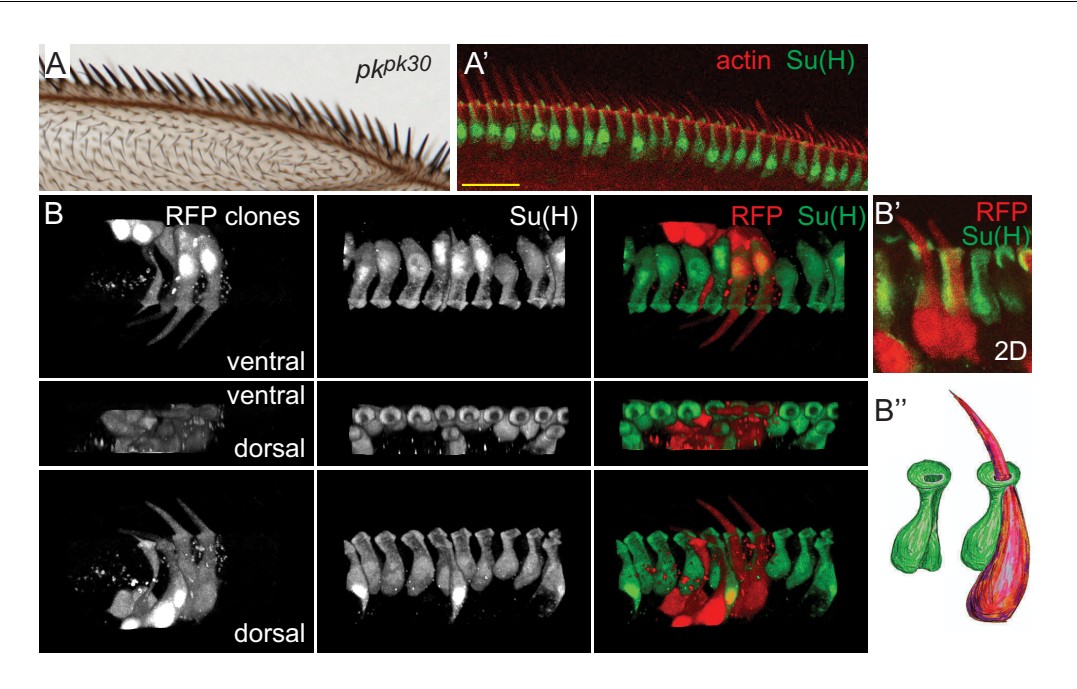

**Figure 2.** Morphology of *pk^pk^* mutant dorsal mechanosensory bristles. (**A–A'**) *pk^pk30^* adult wing and equivalent region of a *pk^pk30^* 36 hr pupal wing stained for Su(H) and actin. (**B–B'**) 3D views of RFP clones in a *pk^pk30^* 36 hr pupal wing revealing reversed orientation of sibling shaft-socket pairs. Su(H) stains socket cells. (**B''**) Cartoon interpretation of shaft-socket pair from B'. Scale bars: 20 μm.
The online version of this article includes the following figure supplement(s) for figure 2:

**Figure supplement 1.** Rose plots for control and *pk^pk30^* socket cell orientations and apical rearrangement of anterior wing margin cells.

nucleus, similar to their arrangement in wildtype, though their alignment is not as regular. Therefore, the structure of *pk^pk^* mutant bristles is roughly a mirror image of that in wildtype, with nuclei in similar positions, but the shaft bending in a counterclockwise direction and rising through a groove on the opposite side of the socket cell compared to wildtype (*Figure 2B–B''*). In the proximal region of the *pk^pk^* mutant wing, the relationship of shaft-socket siblings resembles that in wildtype, reflecting the normal bristle polarity in that region (*Figure 1—figure supplement 1C–C''* and *Figure 3—figure supplement 1B*). Therefore, the P or D position of shaft relative to the sibling socket cell correlates to the normal or reversed bristle polarity in proximal and distal regions of *pk^pk^* mutants, suggesting that the shaft position relative to the socket determines bristle polarity (tilt).

## Helical growth of the shaft-socket pair positions the shaft relative to the socket

Since the relative position of the shaft to the socket appears to be important for bristle polarity, we wished to identify the developmental process by which this relationship is achieved. In microchaete of the notum, the orientation of the initial division of the sensory organ precursor cell is specified by PCP signaling (*Schweisguth, 2015*). Assuming the orientations of subsequent daughter cell divisions are similarly regulated, the shaft-socket relationship may be determined by their relative positions at their birth. A similar process might occur in AWM mechanosensory bristles. Alternatively, a post-division morphogenetic process may determine their final configurations.

To examine this process, we analyzed 3D structures of shaft-socket sibling pairs at earlier developmental stages. In wildtype bristles at 24 hr apf, the shaft cell nucleus is posterior and just slightly dorsal to the socket cell nucleus, similar to their positions at 36 hr apf (*Figure 3A,B*). The extending shaft is just reaching the apical surface, and is positioned in a groove on the dorsal side of the socket cell. At the apical surface, the shaft, sits in a shallow indentation in the apical surface of the socket cell, which adopts a crescent shape with the opening of the crescent pointing in the dorsal direction. Over time, from 28 to 32 to 36 hr apf, as the shaft continues to extend above the apical surface, the

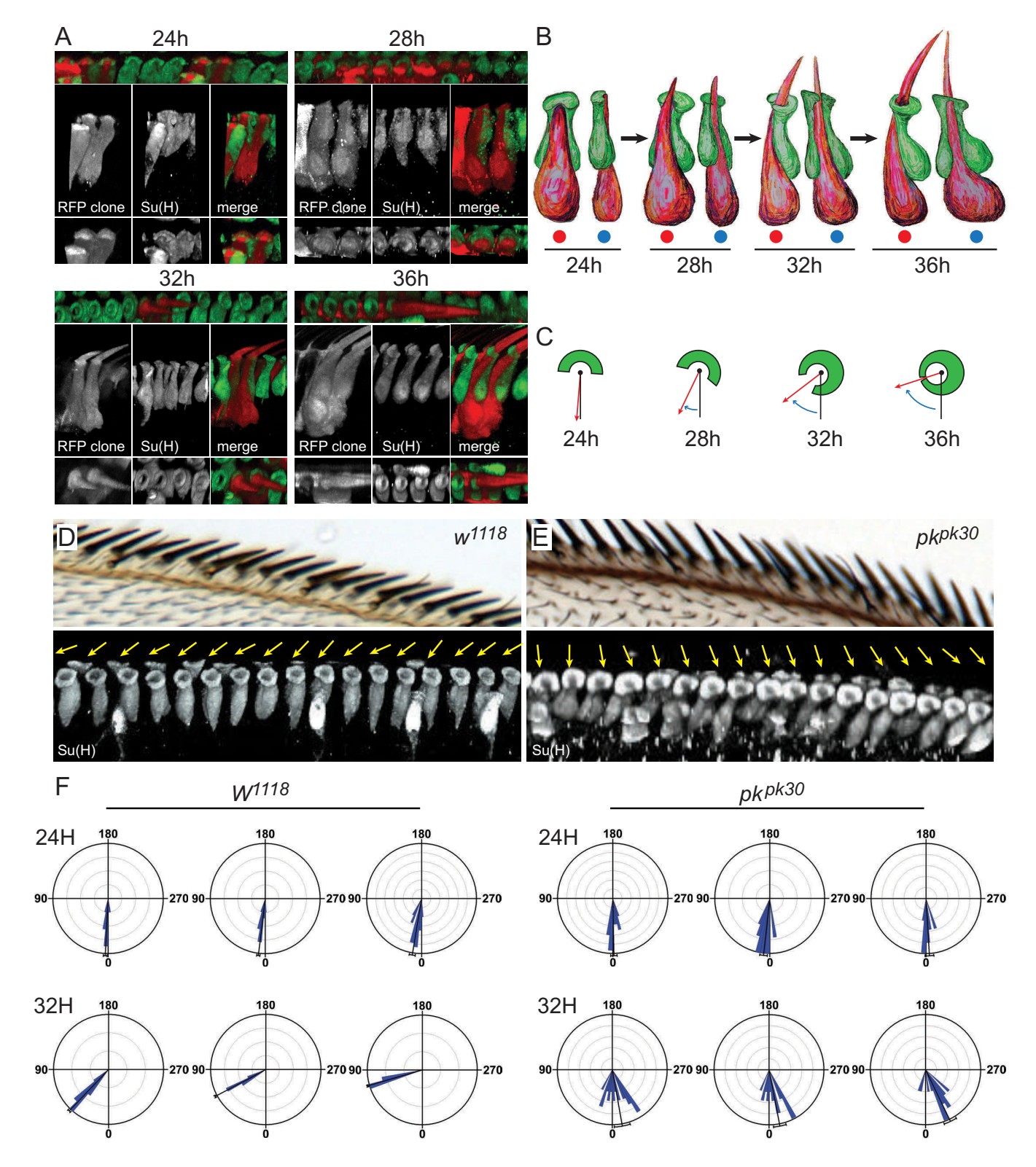

**Figure 3.** 3D images of shaft-socket pairs at different times reveal a clockwise helical growth in control and counterclockwise growth in *pk^pk30* mutant bristles. (**A**) Reconstructed 3D images from varying angles of 24 hr, 28 hr, 32 hr and 36 hr shaft-socket clones marked with RFP and stained with Su(H). (**B**) Cartoon interpretation of images from panel **A**), showing the clockwise rotation of the apical aspects of the shaft and socket cells. Views from dorsal (red dots) and proximal (blue dots). (**C**) Diagrams illustrating scoring of rotation angles. (**D–E**) Control (*w^1118*) and *pk^pk30* adult AWMs and corresponding

*Figure 3 continued on next page*

*Figure 3 continued*

regions from 32 hr pupal wings showing 3D reconstructed images of socket cells stained with Su(H). Orientation angles for these socket cells are indicated by yellow arrows. (F) Quantification of rotation angles for 24 hr and 32 hr control and $pk^{pk30}$ socket cells. Each rose plot represents an individual wing, with scoring limited to the distal AWM anterior to vein L2 unless otherwise indicated (for complete set of rose plots and description of sample sizes, see the legend for *Figure 2—figure supplement 1A*). Most variation between individual wings is likely attributable to variation in developmental timing. For detailed description of quantification, see Materials and methods. Statistical analyses for all genotypes are in *Table 2*.
The online version of this article includes the following video and figure supplement(s) for figure 3:

**Figure supplement 1.** Bristle polarity in adult and socket cell rotation in pupal control ($w^{1118}$), $pk^{pk30}$, $pk^{sple1}$, $pk^{pk-sple13}$, and, *MS1096-Gal4 UAS-pk$^{sple}$* (>>sple) wings at 24 hr and 32 hr apf.

**Figure 3—video 1.** Morphology of 24 hr wildtype dorsal mechanosensory bristles displayed in 3D A reconstruction of confocal stacks displaying AWMs stained with Su(H) (green) to mark socket cells and RFP marking shaft-socket cell pairs.

https://elifesciences.org/articles/51456#fig3video1

**Figure 3—video 2.** Morphology of 28 hr wildtype dorsal mechanosensory bristles displayed in 3D A reconstruction of confocal stacks displaying AWMs stained with Su(H) (green) to mark socket cells and RFP marking shaft-socket cell pairs.

https://elifesciences.org/articles/51456#fig3video2

**Figure 3—video 3.** Morphology of 32 hr wildtype dorsal mechanosensory bristles displayed in 3D A reconstruction of confocal stacks displaying AWMs stained with Su(H) (green) to mark socket cells and RFP marking shaft-socket cell pairs.

https://elifesciences.org/articles/51456#fig3video3

socket cell crescent rotates clockwise, and gradually closes to form a ring around the shaft (*Figure 3A,B*, *Table 2*, *Table 1*, *Figure 3—videos 1*, *2*, *3*). Because the apical surface rotates while the nuclei remain relatively stationary, the shaft and socket twist to form a left-handed helical shape. As the shaft grows out above the socket cell, it points distally. The cell bodies, are initially relatively flat in the dorsal-ventral direction, but as they grow, extend dorsally, becoming flatter in the anterior-posterior direction. Quantification of rotation was performed by measuring rotation angles, as diagrammed in *Figure 3B*, from 3D images of socket cells captured at different time points (c.f. *Figure 3C*), and data are displayed in rose plots (*Figure 3F*, *Figure 2—figure supplement 1A*, *Table 2*). Rotation at 32 hr averages 55° in the clockwise direction. Note that throughout, we describe analyses of right wings. In all cases, left wings develop as mirror images of right wings.

Based on stereotypical patterns from images of timed, fixed samples of the apical surface, we infer that a margin cell at the dorsal side of the shaft-socket pair rotates together with the pair toward the proximal side, generating new junctions with the neighboring socket and margin cells and widening the gap between shaft-socket pairs (*Figure 2—figure supplement 1B*). During these events, the number of margin cells surrounding the shaft-socket pair is maintained and cell-cell junctions are remodeled.

## Counterclockwise helical growth of shaft-socket pairs causes bristle reversal in $pk^{pk}$ mutants

To characterize the events leading to reversed bristle polarity in $pk^{pk}$ mutants, rotation angles of shaft-socket pairs during development were analyzed as above. At 24 hr apf, shaft cells were positioned dorsal to their socket sibling cells, as in wildtype. At later times, apical rotation proceeded counterclockwise, opposite to the wildtype direction, in the distal bristles that adopt a reversed polarity, giving rise to a right-handed helical shape in contrast to the left-handed helical shape of

**Table 1.** p values for comparison of rotation angles for control $W^{1118}$ socket cells at different times apf.

| $W^{1118}$ | 24 hr 5.0° | 28 hr 28.4° |
|---|---|---|
| 24 hr 5.0° | | |
| 28 hr 28.4° | <0.0001 | |
| 32 hr 54.588° | <0.0001 | <0.0001 |

**Table 2.** Summary statistics for rotational angles.
CSD = Circular Standard Deviation.

| Genotype | Time | Grand Mean Vector (GM) | Length of Grand Mean Vector (r) | Number of means (wings) | Mean CSD |
|---|---|---|---|---|---|
| $W^{1118}$ | 24 hr | 5.04° | 0.996 | 6 | 4.14 |
| | 28 | 28.368° | 0.995 | 7 | 4.075 |
| | 32 | 54.588° | 0.981 | 6 | 2.940333 |
| $pk^{pk30}$ | 24 hr | 0.63° | 0.988 | 3 | 7.781333 |
| | 32 hr | 348.482° | 0.946 | 7 | 15.09 |
| $pk^{pk-sple13}$ | 24 hr | 1.061° | 0.997 | 3 | 3.687 |
| | 32 hr | 47.59° | 0.993 | 3 | 6.582 |
| $pk^{sple1}$ | 24 hr | 17.8° | 0.979 | 3 | 7.244667 |
| | 32 hr | 59.288° | 0.99 | 4 | 4.24425 |
| MS1096 >> sple | 24 hr | 358.057° | 0.994 | 3 | 5.796 |
| | 32 hr | 333.296° | 0.964 | 3 | 13.31267 |
| $fz^{R52}$ | 24 hr | 3.351° | 0.998 | 3 | 3.923333 |
| | 32 hr | 20.471° | 0.905 | 6 | 23.5668 |
| $dsh^1$ | 24 hr | 5.684° | 0.996 | 2 | 5.023 |
| | 32 hr | 20.144° | 0.944 | 6 | 14.316 |
| fmi RNAi | 24 hr | 1.15° | 0.996 | 3 | 4.78 |
| | 32 hr | 19.291° | 0.978 | 4 | 11.741 |
| $vang^{stbm6}$ | 24 hr | 1.51° | 0.967 | 3 | 11.52167 |
| | 32 hr | 25.447° | 0.94 | 6 | 14.21117 |
| $w^{1118}$ MS1096 > dsRNAi | 32 hr | 16.593° | 0.79 | 4 | 33.10867 |
| $pk^{pk}$ MS1096 > dsRNAi | 32 hr | 29.323° | 0.961 | 3 | 14.37067 |
| $pk^{sple}$ MS1096 > dsRNAi | 32 hr | 6.596° | 0.97 | 3 | 12.63867 |
| $pk^{pk-sple}$ MS1096 > dsRNAi | 32 hr | 40.297° | 0.99 | 3 | 6.219667 |

*Table 2—source datas* **1**, **2**, **3**, **4**, **5**, **6**, **7**, **8**, **9**, **10**, **11**, **12**, **13**, containing measurement of rotation angles of socket cells in control and various PCP mutant wings, with associated statistics, are provided.

The online version of this article includes the following source data for Table 2:

**Source data 1.** Source data for $w^{1118}$.

**Source data 2.** Source data for $pk^{30}$.

**Source data 3.** Source data for $pk^{pk-sple13}$.

**Source data 4.** Source data for $pk^{sple1}$.

**Source data 5.** Source data for *MS1096-GAL4; UAS-pk$^{sple}$*.

**Source data 6.** Source data for $fz^{R52}$.

**Source data 7.** Source data for $dsh^1$.

**Source data 8.** Source data for *fmiRNAi*.

**Source data 9.** Source data for $vang^{stbm6}$.

**Source data 10.** Source data for $w^{1118}$; *MS1096-GAL4; UAS-dsRNAi*.

**Source data 11.** Source data for $pk^{pk}$; *MS1096-GAL4; UAS-dsRNAi*.

**Source data 12.** Source data for $pk^{sple}$; *MS1096-GAL4; UAS-dsRNAi*.

**Source data 13.** Source data for $pk^{pk-sple}$; *MS1096-GAL4; UAS-dsRNAi*.

wildtype pairs (*Figure 3E*, *Figure 2—figure supplement 1A*, *Table 3*). Shaft-socket pairs in the proximal region rotated clockwise, corresponding to their normal polarity, and bristles in the region between normal and reversed bristles rotated very little, corresponding to their neutral, upright, polarity (*Figure 3—figure supplement 1B*). Therefore, bristle polarity does not depend on the birth

positions of shaft and socket cells, which are born by 14 hr apf (*Hartenstein and Posakony, 1989*), but rather on the direction of apical rotation of the developing shaft-socket pair, beginning shortly after 24 hr apf.

## Pk^pk versus Pk^sple isoform expression determines the direction of rotation

Since $pk^{pk}$ mutants show counterclockwise rotation leading to reversed shaft-socket positioning (D-P) and bristle reversal, and Pk^sple overexpression similarly reverses bristle polarity as previously reported (*Figure 3—figure supplement 1E,E'* and *Gubb et al., 1999*), we surmised that Pk^sple induces the counterclockwise helical growth that leads to D-P orientation of shaft-socket pairs and reversed bristle polarity. Consistent with this, counterclockwise rotation of shaft-socket pairs and proximal polarity in $pk^{pk}$ mutants is suppressed by removing $pk^{sple}$ (in $pk^{pk-sple13/pk-sple13}$; *Figure 3— figure supplement 1A–D'*, *Table 3*, *Gubb et al., 1999*), and $pk^{sple}$ overexpression induces counterclockwise rotation similar to that in $pk^{pk}$ mutants (*Figure 3—figure supplement 1E,E' Table 2*, *Table 3*). Pk^sple is therefore needed to reverse shaft-socket rotation in $pk^{pk}$ mutants. Overexpression of Pk^sple induced counterclockwise rotation and reversed polarity in a wider region than in $pk^{pk}$ mutants (compare *Figure 3—figure supplement 1B* to E). Thus, endogenous Pk^sple is only poised to act at the distal margin, but exogenous Pk^sple can reverse most, if not all, bristles. Though the potential for Pk^sple to reverse bristle polarity is unmasked in the absence of Pk^pk, it plays no essential role in wildtype polarization, as $pk^{sple}$ bristles fully rotate, or perhaps marginally over-rotate (59.3°±4.2° vs 54.6°±2.9°, p=0.0702; *Table 3*, *Figure 3—figure supplement 1C,C'*).

Surprisingly, $pk^{pk-sple13}$ mutant bristles rotate only moderately less than wildtype bristles (47.6°± 6.6° vs. 54.6°±2.9°, p=0.0143; *Table 3*), suggesting that Pk^pk might play only a modest role in controlling the magnitude of rotation in wildtype. We propose that this is due to residual core PCP signaling activity observed in the absence of Pk (*Strutt and Strutt, 2007*; *Lawrence et al., 2004*; *Adler et al., 2000*), and the implications of this result are considered more fully in the Discussion.

## Localization of Pk^pk and/or Pk^sple correlates with handedness of helical rotation

We have shown that in wildtype bristles, Pk^pk antagonizes Pk^sple to direct clockwise rotation of shaft-socket pairs, and that Pk^sple, when overexpressed, outcompetes Pk^pk to direct counterclockwise rotation. Although the idea that Pk^pk and Pk^sple antagonize each other has been previously proposed (*Gubb et al., 1999*), how this occurs has been obscured in part by the inability to specifically visualize the endogenous expression of each isoform. We therefore modified the endogenous genomic sequence encoding Pk^pk or Pk^sple by appending a V5 tag to the N-terminus, facilitating the tissue and cellular level evaluation of their native expression patterns at various developmental stages. Both tagged isoforms support wildtype polarity development in all tissues and various controls suggest that expression of these genomically tagged versions reflect that of the native loci (*Figure 4*, *Figure 4—figure supplements 1* and *2*). Here, we describe their expression in developing wings.

**Table 3.** p values for comparison of rotation angles for control and *pk*-related genotypes at 32 hr apf. ns = not significant.

| 32 hr angles | $W^{1118}$ 54.6° | $pk^{sple1}$ 59.3° | $pk^{pk-sple13}$ 47.6° | $pk^{pk30}$ 348.5° |
|---|---|---|---|---|
| $W^{1118}$ 54.6° | | | | |
| $pk^{sple1}$ 59.3° | (ns) 0.0702 | | | |
| $pk^{pk-sple13}$ 47.6° | 0.0143 | 0.0343 | | |
| $pk^{pk30}$ 348.5° | <0.0001 | <0.0001 | 0.0004 | |
| >>Pksple 333.2° | 0.0012 | <0.0001 | 0.001 | (ns) 0.1826 |

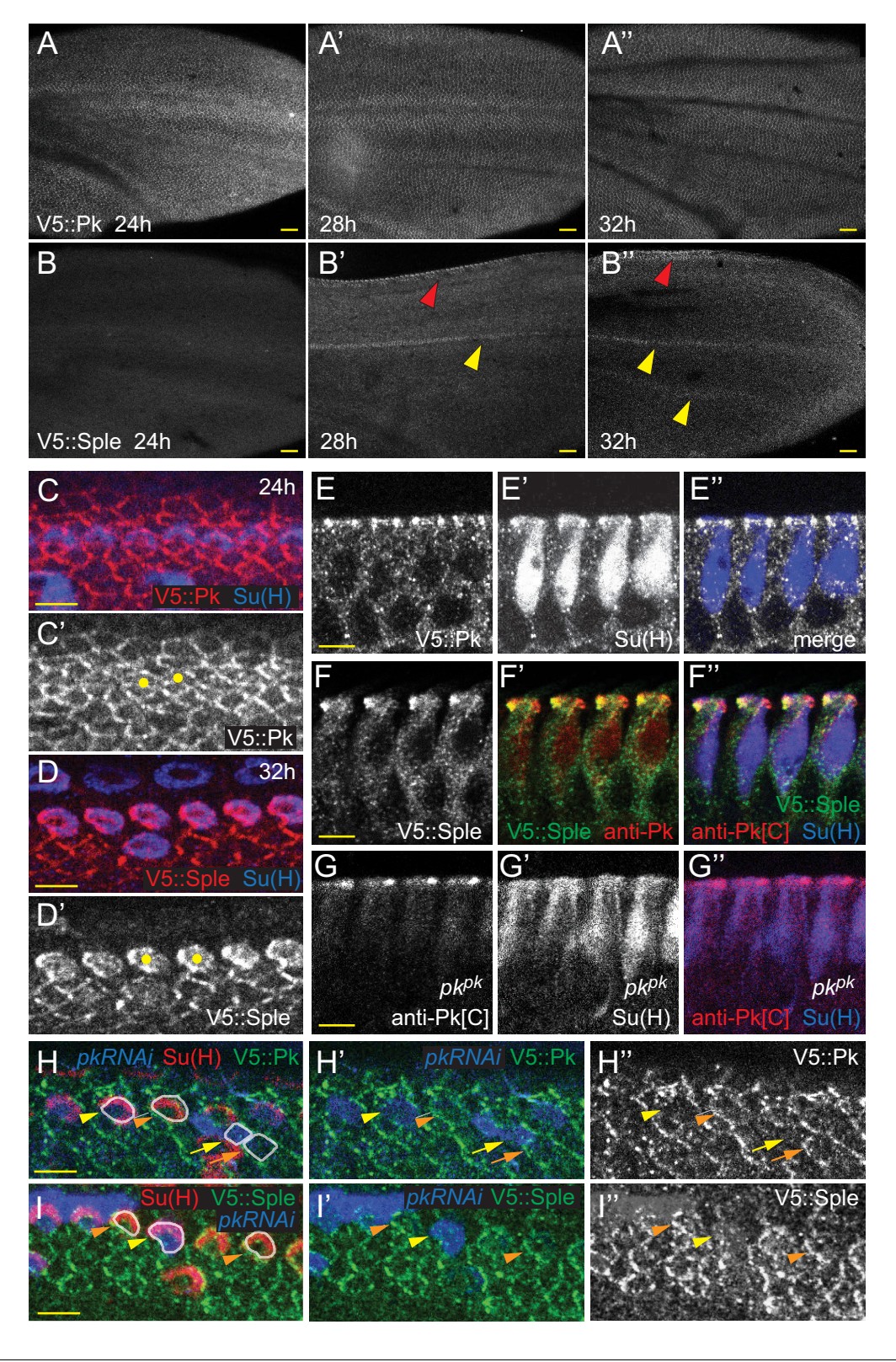

**Figure 4.** Expression of Pk[pk] and Pk[sple] isoforms in pupal wings. (A–B'') V5::Pk (A–A'') and V5::Sple (B–B'') in pupal wings of ages indicated. Red arrowheads mark AWM and yellow arrowheads mark veins L3 and L4. (C–D') Surface views of V5::Pk at 24 hr (C–C') or V5::Sple at 32 hr (D–D') counterstained for Su(H) to locate socket cells. Some

*Figure 4 continued on next page*

*Figure 4 continued*

socket cell locations are indicated by yellow dots. (**E–G''**) Planar sections of socket cells (Su(H)) of 28 hr pupal wings with apical at the top. V5::Pk is at all junctions between socket and margin cells (**E–E''**). Pk$^{sple}$ (detected with V5::Sple) appears to be localized to the proximal side of control ($w^{1118}$) socket cells (**F–F''**) but to the distal side of $pk^{pk}$ mutant socket cells (detected with anti-Pk[C]; **G–G''**). (**H–I'**) Mosaic expression of V5::Pk (**H–H''**) or V5::Sple (**I–I''**) in otherwise wildtype wings (28 hr). Cells lacking expression are marked with RFP (blue). Both V5::Pk and V5::Sple localize to the proximal side of expressing (orange arrowheads) but not non-expressing (yellow arrowheads) socket cells, demonstrating their proximal localization. V5::Pk localizes to the proximal side of expressing (orange arrow) but not non-expressing (yellow arrow) margin wing cells. V5::Sple is also proximal in margin wing cells, though no informative clones were captured in this image. Several relevant cells are outlined in (**H and I**) for clarity. Scale bars: 20 µm (**A,B**) and 5 µm (**C–I**).

The online version of this article includes the following figure supplement(s) for figure 4:

**Figure supplement 1.** V5::Pk and V5::Sple expression throughout wing development.
**Figure supplement 2.** Timing of expression and localization of Pk$^{pk}$ and/or Pk$^{sple}$ correlates with handedness of helical rotation.

Notably, V5::Pk and V5::Sple reveal that throughout wing development, expression is spatiotemporally dynamic. Early in wing development, Pk$^{pk}$ protein is strongly expressed and is present in most or all cells. In discs, Pk$^{pk}$ is relatively elevated in AWM proneural cells. Following a slight dip in levels around 8 hr apf, expression levels climb, peaking around 32 hr apf, when wing hairs emerge, and then decline with little detectable expression remaining by 40 hr apf (***Figure 4A–A''***, ***Figure 4—figure supplement 1B–H,K–K''***). Expression of Pk$^{sple}$ is below detection in discs (***Figure 4—figure supplement 1B,I,J***), makes a small peak at around 8 hr apf, and is then not detectable between 16 and 28 hr apf. Beginning around 28 hr apf, Pk$^{sple}$ expression is specifically detected in dorsal AWM cells and in vein L3. This expression persists through 32 hr apf, when wing hairs emerge, and weaker expression in other veins becomes apparent (***Figure 4B–B''***, ***Figure 4—figure supplement 1B,L,L'***). At these times, Pk$^{sple}$ is below detection levels on Western blots. Beginning sometime after 32 hr apf, Pk$^{sple}$ expression increases in most cells, with its level equaling that of the declining Pk$^{pk}$ by 36 hr apf and reaching its highest level at around 40 hr apf (the latest time we examined), when Pk$^{pk}$ is no longer detected. At 40 hr apf, when Pk$^{sple}$ is at peak expression in most of the wing blade, expression has disappeared from vein cells (***Figure 4—figure supplement 1B,L–L''***).

Most pertinent to bristle development, at the 24 hr apf AWM, Pk$^{pk}$ is expressed at apical junctions at similar levels in anterior margin and socket cells (***Figure 4C,C'***, ***Figure 4—figure supplement 2A***). By 28 hr apf, Pk$^{pk}$ expression begins to decrease during bristle-socket rotation, first in socket cells, and later in all cells at and near the margin (***Figure 4—figure supplement 2A***). At the same time, Pk$^{sple}$ expression becomes evident and increases over time, first uniformly in cells near the margin, and gradually becoming strongest in the socket and shaft cells (***Figure 4D,D',F–F''***, ***Figure 4—figure supplement 2B***, compare with Pk$^{pk}$ (V5::Pk) in ***Figure 4E***). The timing of the shift from Pk$^{pk}$ to Pk$^{sple}$ expression is accelerated at the margin relative to the interior of the wing. Mosaic experiments demonstrate that Pk$^{pk}$ localizes proximally in socket and other margin cells, and that, importantly, Pk$^{sple}$ colocalizes with Pk$^{pk}$ to the proximal side of socket cells. Proximal Pk$^{sple}$ localization is an unexpected observation based on previous studies in which overexpressed Pk$^{sple}$ localized at the distal junctions of hair cells (***Ayukawa et al., 2014***; ***Ambegaonkar and Irvine, 2015***). Because Pk$^{sple}$ expression is stronger in socket cells compared to margin cells by 32 hr apf, it has the useful property of effectively being expressed as a mosaic, allowing its localization to be scored without inducing clones (***Figure 4F–F'',H–H'',I–I''***).

As Pk$^{sple}$ activity reverses rotation direction of shaft-socket pairs in $pk^{pk}$ mutant wings, Pk$^{sple}$ protein localization was analyzed in $pk^{pk}$ mutant wings. Because anti-Pk[C] antibody recognizes the common region of Pk$^{pk}$ and Pk$^{sple}$, the antibody reveals Pk$^{sple}$ isoform localization in $pk^{pk}$ mutants. In the region of $pk^{pk}$ mutant wings where P-D reversal occurs, Pk$^{sple}$ protein localized at the distal side of socket cells, whereas in the proximal region where polarity is not reversed, Pk$^{sple}$ protein shows minimal asymmetry (***Figure 4G–G''***, ***Figure 4—figure supplement 2C–C'''***). Distal localization of Pk$^{sple}$ in the region of polarity reversal was verified by clonal knockdown of $pk^{sple}$ in $pk^{pk}$ mutant wings. Notably, the socket-shaft pairs that lacked both Pk$^{sple}$ and Pk$^{pk}$ failed to rotate (***Figure 4—figure supplement 2D–E'''***).

These results suggest that Pk$^{pk}$ normally inhibits Pk$^{sple}$ from localizing distally by recruiting it to the proximal junction of socket cells (and likely also in nearby margin cells, although this is hard to visualize due to lower expression in those cells). Furthermore, the distal localization of Pk$^{sple}$ in $pk^{pk}$ mutants correlates with its ability to determine counter-clockwise rotation of shaft-socket pairs on a cell-by-cell basis, suggesting that this localization is likely the determinant of counter-clockwise rotation.

## Core PCP components control rotation of shaft-socket pairs

Our results thus far show that the direction of Pk polarization, whether Pk$^{pk}$, Pk$^{sple}$ or both, corresponds to the direction of bristle polarization. Suppression of polarity reversal in $pk^{pk}$ mutants by *dsh* implicates the core PCP signaling mechanism in this process *Gubb et al. (1999)*. We therefore asked whether the remaining components of the core PCP signaling mechanism contribute to AWM bristle polarization. As with *dsh* mutation, knock down of *fz* or *vang* in $pk^{pk}$ mutants suppressed reversal of bristle tilt, shaft-socket orientation, and rotation direction (*Figure 5A–D*). Core PCP signaling is therefore required for reversed, Pk$^{sple}$-dependent polarity.

To assess a potential contribution of core PCP signaling to Pk$^{pk}$-dependent bristle polarization, the anterior region of adult wings from *fmi* RNAi, *fz*, *vang*, and *dsh* mutants were analyzed, and the rotation of shaft-socket pairs was evaluated for each genotype (*Figure 5E–H*). Adult mechanosensory bristles of core PCP mutants are less tilted toward the distal direction than those of wildtype and the tilting angles are somewhat irregular, with some bristles tilting out of the plane of the wing. Consistent with the adult wing defects, in pupal wings the shaft position relative to the socket varies, sometimes abruptly, in the same mutant wing, showing less local correlation than in wildtype. Quantification reveals substantial under-rotation of shaft-socket pairs, and a broader distribution of rotation angles than in wild type (*Figure 5E–H*, compare with *Figure 3D,F*, *Table 2*). Thus, careful morphological analysis reveals that core PCP signaling is required for normal, Pk$^{pk}$-dependent polarity as well as reversed, Pk$^{sple}$-dependent polarity.

Consistent with a role for core PCP components in mediating rotation of shaft-socket pairs, junctional asymmetry of Fz::EGFP and Vang::EYFP is well preserved between socket and margin cells and between adjacent margin cells (*Figure 5I–J'*). Little accumulation was observed at junctions between shaft and socket cells. Mosaic analyses demonstrated the expected distal localization of Fz and proximal localization of Vang at both margin cell-margin cell and margin cell-socket cell junctions, suggesting that PCP signaling likely occurs between margin cells and between margin and socket cells (*Figure 5K,L*). Similarly, the reversed bristle polarity observed in $pk^{pk}$ mutants was accompanied by reversed Vang localization in $pk^{pk}$ mutant socket and margin cells, consistent with the idea that the direction of core PCP polarization is reversed in $pk^{pk}$ mutants (*Figure 5—figure supplement 1*). To functionally test polarity propagation between margin and socket cells by core PCP components, *fz* or *vang* knock-down clones were generated, and clones at the AWM were analyzed (*Figure 6*, *Figure 6—figure supplement 1*). *fz* or *vang* knock-down clones, whether in just margin cells, just shaft-socket pairs, or both, showed non-autonomy as assessed by sequestration of Fz (*fz* clones) or Vang (*vang* clones) at the clone borders. Near *fz* RNAi clones, distal cells, including both bristle and hair cells, were re-oriented: sockets on the distal side of the clones showed counter-clockwise rotation, and hairs on the distal side grew toward the clones. Similarly, sockets on the proximal side of *vang* RNAi clones rotated counter-clockwise (*Figure 6—figure supplement 1*). These results indicate that core PCP signaling propagates between bristle and margin cells to control the rotation direction of shaft-socket pairs.

## The core PCP module differentially interprets directional signals from ds when operating in Pk$^{pk}$- or Pk$^{sple}$-dependent modes

We have previously proposed that a signal from the Ft/Ds/Fj system provides a directional cue to orient core PCP signaling in some tissues (*Ma et al., 2003*; *Yang et al., 2002*; *Matis et al., 2014*; *Olofsson et al., 2014*), although others have argued that this system operates in parallel with core PCP signaling (*Casal et al., 2006*; *Lawrence et al., 2007*; *Brittle et al., 2012*). An asymmetry of Ft-Ds heterodimers, with a small excess of Ds displayed on the distal side of the cell, and Ft on the proximal side, has been observed, and is proposed to provide this signal (*Ambegaonkar et al., 2012*; *Bosveld et al., 2012*; *Brittle et al., 2012*). We have also proposed that the core PCP module

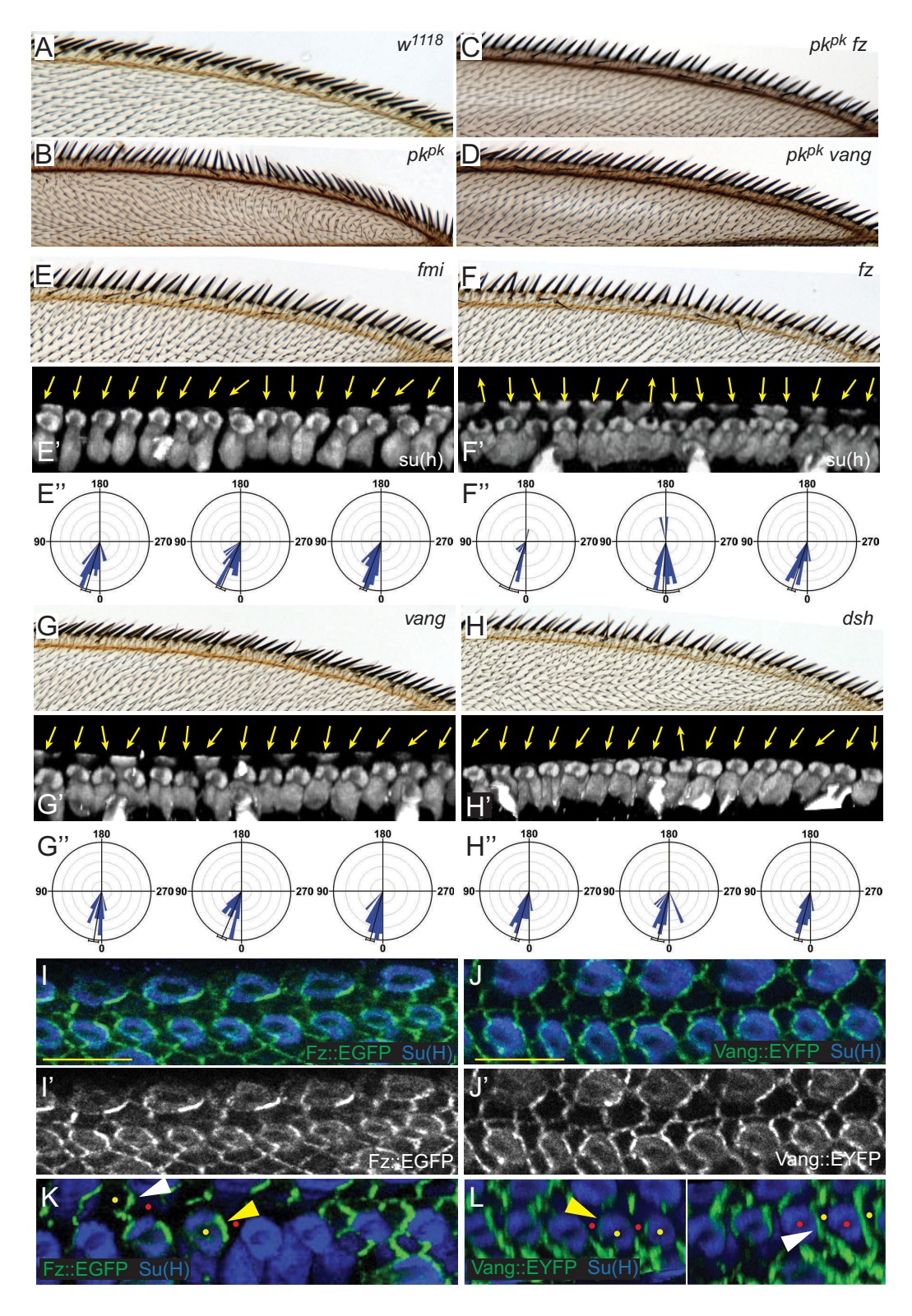

**Figure 5.** Core PCP components control shaft-socket rotation. (A–D) Reversed bristle polarity in *pk^pk^* mutant compared to control is abrogated in *pk fz* and *pk vang* double mutants, demonstrating requirement for core PCP activity for polarity reversal in *pk^pk^* mutants (reversed bristle polarity is suppressed in all double mutant wings analyzed (n = 20 of each genotype)).(E–H) While bristle tilt is only mildly disturbed, socket cell rotation is strongly impaired in *fmi* knockdown, *fz, vang* and *dsh* mutant wings. Adult wings, representative socked cell images (32 hr) and quantification of three

*Figure 5 continued on next page*

*Figure 5 continued*

individual wings (32 hr) for each genotype are shown (*fmi* knockdown - 97 sockets from four wings; *fz* mutant - 120 sockets from six wings; *vang* muant - 114 sockets from six wings; and *dsh* mutant - 110 sockets from six wings). Statistical analyses for all genotypes are in *Table 2*. (I–J') Expression of Fz:: EGFP (I–I') and Vang::EYFP (J–J') in margin cells. (K–L) Mosaic expression demonstrates distal localization of Fz::EGFP (K) and proximal localization of Vang::EYFP (L) in both socket (yellow) and margin (white arrowheads) cells. Some informative expressing cells (yellow dots) next to non-expressing cells (red dots) are marked. Scale bars: 10 μm. Statistical analyses for all genotypes are in *Table 2*.

The online version of this article includes the following figure supplement(s) for figure 5:

**Figure supplement 1.** Reversed Vang localization in *pk^pk* mutant socket and margin cells.

differentially interprets directional signals from Ds when operating in Pk^pk- or Pk^sple-dependent modes, with Pk^sple directing localization of the Fmi-Vang complex to the side where Ds is in excess, while the Fmi-Vang complex localizes to the opposite side when functioning in a Pk^pk-dependent manner [(*Olofsson and Axelrod, 2014*); see also *Lawrence et al. (2004)*. Pk^sple has been shown to bind to Ds and the associated Dachs protein, providing a mechanism for orienting Pk^sple-dependent core function to the Ft/Ds/Fj signal (*Ayukawa et al., 2014*; *Ambegaonkar and Irvine, 2015*), whereas a less direct, microtubule-dependent mechanism was proposed to mediate this response when Pk^pk is predominant (*Shimada et al., 2006*; *Harumoto et al., 2010*; *Olofsson and Axelrod, 2014*; *Matis et al., 2014*). In contrast, some have proposed that Pk^pk-dependent core signaling is instead uncoupled from the Ft/Ds/Fj signal (*Merkel et al., 2014*; *Ambegaonkar and Irvine, 2015*).

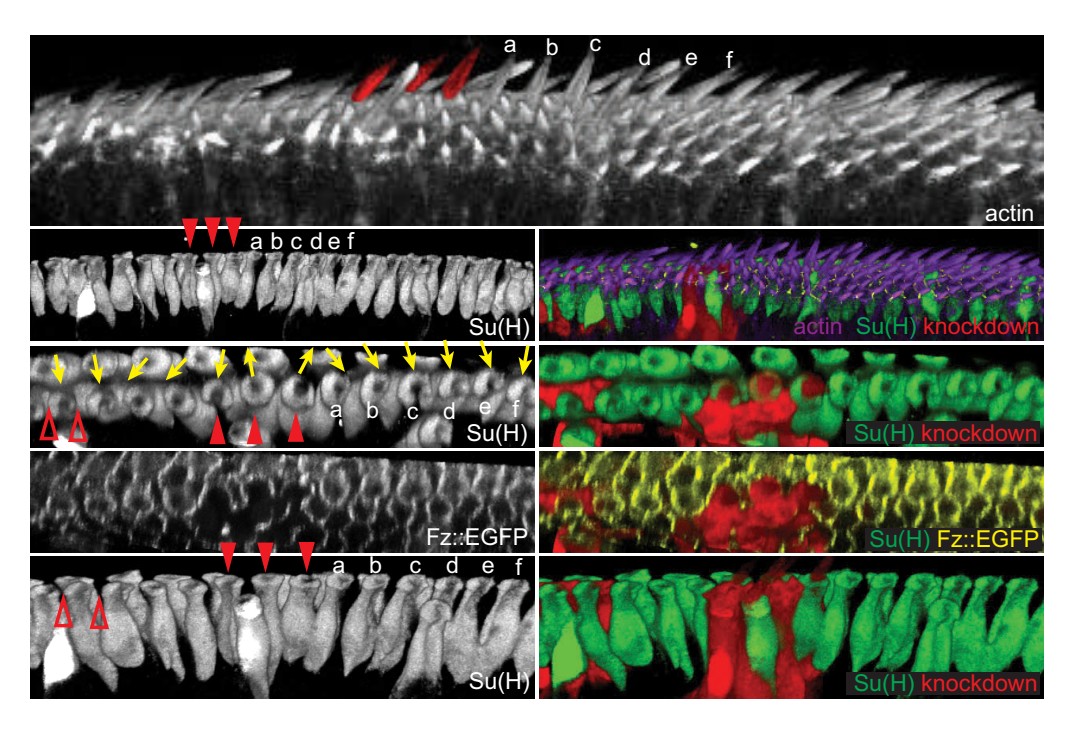

**Figure 6.** Polarity propagates between bristle and margin cells. 3D reconstructed views of a Fz::EGFP wing (32 hr) with *fz* knockdown clones, stained for actin to mark wing hairs and bristle shafts, Su(H) to mark socket cells and RFP to indicate knockdown clones. A clone involving three bristles (red arrowheads, false-colored in top image) shows domineering non-autonomy, reversing the polarity of nearby bristles and hairs on the distal side of the clone (bristles marked a-f). The effect on both hair cells and bristles diminishes with distance. A small clone affecting margin cells (red open arrowheads) disrupts the polarity of bristle cells on the distal side of the mutant cells. 20 *fz* RNAi clones from eight wings were analyzed for cell autonomous and non-autonomous effects. All clones showed cell autonomous polarity disruption and non-autonomous reversal of varying extent depending on the clone size.

The online version of this article includes the following figure supplement(s) for figure 6:

**Figure supplement 1.** *vang*RNAi clone showing polarity propagation among bristles at the margin.

We reasoned that our reagents would allow us to analyze effects of the Ft/Ds/Fj signal on each Pk isoform. We first examined the phenotype resulting from knockdown of Ds (*Figure 7C*). As previously observed (*Adler et al., 1998*), normal bristle tilt was substantially disturbed. The pattern of disturbance consistently showed regions of coordinated polarity that smoothly transition through neutral polarity to adjacent regions of opposite polarity, although the number and position of those domains varied. The effectively mosaic expression of V5::Sple localization allows one to observe precisely correlated regions of distal and proximal Sple localization corresponding to the regions of reversed and normal polarity, respectively, with relatively symmetric localization in the intervening transitions (*Figure 7C*). These results suggest that local core PCP signaling maintains polarity correlation among immediate neighbors but that alignment to the tissue axis is eliminated in the absence of Ds.

We then asked if the Pk$^{sple}$-dependent reversed polarity in $pk^{pk}$ mutants depends on Ds. When *ds* was knocked down in $pk^{pk}$ mutants, bristle reversal was blocked, producing a phenotype similar to that of core mutants, and the distal localization of Pk$^{sple}$ was no longer observed (*Figure 7A–D*). Therefore, the reversed, Pk$^{sple}$-dependent polarity in $pk^{pk}$ mutants requires Ds, and we interpret this to indicate that the Ds global signal recruits Pk$^{sple}$ to sites of enriched Ds (distal) in the absence of Pk$^{pk}$ (*Figure 7B*; compare with 7D), which drives reversal of shaft-socket orientation and therefore reversal of bristle polarity.

In *ds* knockdowns, we are unable to readily interpret the localization pattern of V5::Pk because levels are similar in socket and margin cells. Nonetheless, other results suggest that the local polarity correlation in the absence of Ds is mediated primarily by asymmetric localization of Pk$^{pk}$ rather than the asymmetric localization of Pk$^{sple}$ that we can observe. First, recall that in wildtype, Pk$^{sple}$ is recruited to colocalize with Pk$^{pk}$ at proximal sites, so Pk$^{pk}$ is likely to similarly recruit the colocalization of Pk$^{sple}$ in the absence of Ds. Consistent with this idea, when *ds* was knocked down in $pk^{pk}$ mutants, neither proximal nor distal localization of Pk$^{sple}$ was observed in socket cells, and local correlation of bristle polarity was weak (*Figure 7D*). Thus, the local domains of correlated asymmetric Pk$^{sple}$ localization in *ds* knock-down socket cells depend on the presence of Pk$^{pk}$; Pk$^{sple}$ alone is insufficient to facilitate local signaling between neighbors. Finally, removing Pk$^{sple}$ in *ds* knock-down wings failed to significantly modify *ds* knock-down effects on the polarity of bristles (and also hairs) while removing both Pk$^{pk}$ and Pk$^{sple}$ does (*Figure 7E,F*, *Figure 7—figure supplement 1A–D*), confirming that *ds* knock-down affects the Pk$^{pk}$-mediated, rather than the Pk$^{sple}$-mediated, PCP signal for wing bristle (and hair) polarity. Taken together, these observations suggest that the locally correlated domains of polarity observed in *ds* knock-down wings depend on Pk$^{pk}$ activity.

These and previous results indicate that Pk$^{pk}$ is the principal isoform functioning in core PCP signaling during bristle polarization. They are most consistent with, though do not definitively show, that in wildtype, the Ds global signal directs orientation of core signaling such that Vang and Pk$^{pk}$ localize to the proximal side (and incidentally colocalizing Pk$^{sple}$ to the proximal side) to establish normal polarity. The alternative possibility is that Ds activity is permissive, and some other signal directs this orientation of Pk$^{pk}$-dependent core polarization. The proposal that Ds is instructive for orienting Pk$^{pk}$-dependent core signaling, while consistent with our prior interpretation of coupling between the Ft/Ds/Fj signal and Pk$^{pk}$-dependent core PCP signaling in wing hair polarization (*Ma et al., 2003*; *Olofsson et al., 2014*; *Sharp and Axelrod, 2016*; *Yang et al., 2002*), is at odds with other reports asserting that while under Pk$^{pk}$ control, core PCP directionality is uncoupled from the Ft/Ds/Fj signal (*Merkel et al., 2014*; *Ambegaonkar and Irvine, 2015*). Rigorous testing of this hypothesis requires reorienting the Ft/Ds/Fj signal and assessing the isoform dependence of the response.

It was previously shown that reversing the gradient of Ds expression near the distal part of the wing under control of *distal-less-GAL4* (*dll >2* x-ds) reverses wing hair polarity (*Harumoto et al., 2010*). Assuming that hair polarity is determined by Pk$^{pk}$, for which ample evidence exists, and that it depends on core signaling, this result would demonstrate coupling of Pk$^{pk}$-dependent core PCP signaling to the Ds signal. We rigorously tested this assumption by testing the core signaling and Pk isoform dependence of this response (*Figure 7G–K*). *dll >2* x-ds reverses polarity of a substantial swath of wing hairs, precisely in the region where the Dll expression gradient is expected to be steepest (*Figure 7—figure supplement 1E*). *dll >2* x-ds, however does not reverse AWM bristle polarity, as it does not produce a proximal-to-distal expression gradient at the AWM. Because our results show that AWM bristle and wing hair polarization show indistinguishable responses to Ft/Ds

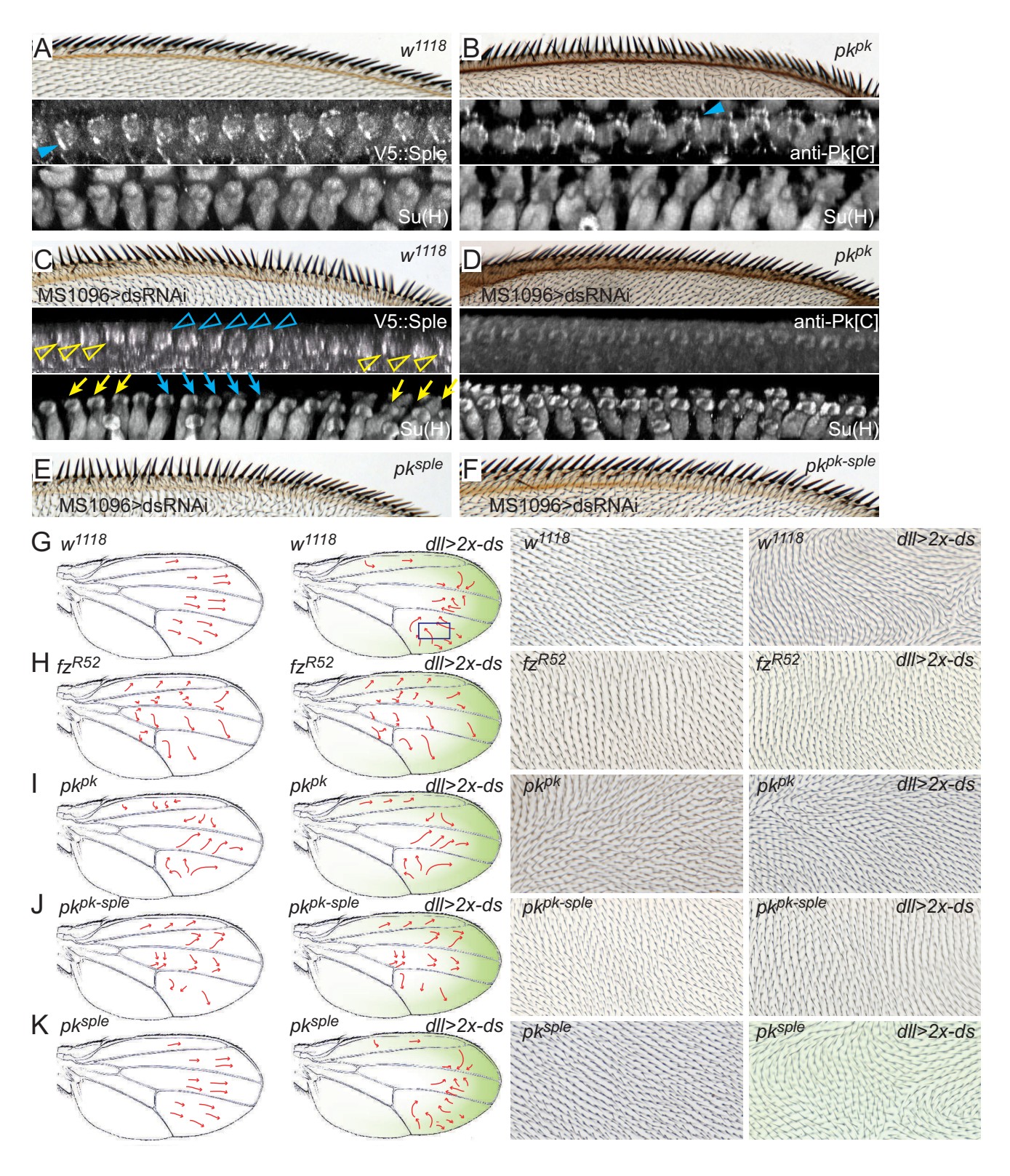

**Figure 7.** Pk[pk] and Pk[sple] activity responds to Ds. (**A**) A control (*w[1118]*) wing showing normally oriented bristles and rotated socket cells, and proximal V5::Sple. (**B**) A *pk[pk]* mutant wing with reversal of distal bristles, counter-rotated socket cells and distal Pk[sple]. (**C**) Wings knocked down for *ds* (*MS1096-GAL4, UAS-ds[RNAi]*) show variable regions of locally correlated but either reversed or normal polarity. V5::Sple localization varies, corresponding to the

*Figure 7 continued on next page*

*Figure 7 continued*

local reversed (blue) or normal (yellow) polarity. (**D**) In *pk^pk^* mutant wings in which *ds* is knocked down, socket cells are minimally rotated, and Pk^sple^, probed with anti-Pk[C] antibody, shows minimal apical localization. (**E**) *pk^sple^* wing with *ds* knocked down. (**F**) *pk^pk-sple^* wing with *ds* knocked down. Quantification of socket cell rotation for the genotypes shown in **C–F** are given in ***Figure 7—figure supplement 1***. (**G–K**) The normal proximal-high and distal-low Ds gradient was reversed in the distal wing by driving two copies of *UAS-ds* by *dll-Gal4* in control (*w^1118^*), *fz^R52/R52^*, *pk^pk^*, *pk^pk-sple^* and *pk^sple^* wings. The approximate gradient of ectopic ds expression is shown in green. Polarity reversal in the distal part of a control wing (**G**) was reversed (arrows, and images representing boxed regions) and depended on the presence of Fz (**H**) and Pk^pk^ but not on Pk^sple^ (**I–K**). Wing hair images (**G–K**) are of the dorsal side. Subjectively similar hair polarity patterns were obtained from ≥13 of 15 wings for each genotype in G-K.
The online version of this article includes the following figure supplement(s) for figure 7:

**Figure supplement 1.** A reversed ectopic Ds gradient re-orients core PCP domains.

and to core PCP manipulations, we propose that wing hairs are a suitable readout for this assay. We first asked whether *dll >2 x-ds*-driven hair polarity reversal depends on core module activity by removing Fz, and found that ectopic Ds-dependent reversal is blocked in a *fz* mutant background (***Figure 7G,H***). Furthermore, ectopic Ds re-orients the core PCP protein orientation (***Figure 7—figure supplement 1F–I***). These results rule out the possibility that ectopic Ds reverses polarity through a pathway that does not include the core PCP module. We then tested the Pk isoform dependence of reversal, and found that it is almost entirely abolished upon removal of Pk^pk^ (*pk^pk^* or *pk^pk-sple^*), but is largely unchanged upon removal of Pk^sple^ (*pk^sple^*) (***Figure 7G,I–K***). This result decisively demonstrates that Pk^pk^-dependent core PCP signaling in wing hair polarization is oriented by the Ft/Ds/Fj signal, and strongly suggests that the same coupling occurs during Pk^pk^-dependent AWM bristle polarization.

## Discussion

### Direction of helical morphogenesis determines bristle polarity

Producing structures of defined chirality requires directional information on three Cartesian axes. Our results indicate that in determining bristle chirality, PCP provides directional information along the proximal-distal axis. The apical-basal axis is defined by the epithelium, while the dorsal-ventral axis is likely defined by the dorsal-ventral compartment boundary.

We have shown that the polarity of wing margin bristles (proximal or distal tilt) is determined by controlling the handedness of helical growth. The entwined shaft and socket cells undergo an apical clockwise or counterclockwise rotation that results in a left-handed or right-handed helical structure, placing the shaft to the proximal (wildtype) or distal (*pk^pk^* mutant) side of the socket cell. The direction of rotation depends on PCP signaling among and between margin and socket cells. Helical cellular structures of defined handedness, such as the bristles resulting from properly directed rotation, have been noted in bacterial and plant species, but few examples have been described in animals.

The entwined twisting of the shaft and socket is a coordinated morphogenetic event, and the apparent stereotyped junctional rearrangement of additional margin cells suggests that at least some other cells are involved as well. We do not know in which cell or cells mechanics are regulated to drive this morphogenesis. One possibility is that an internal cytoskeletal mechanism induces the helical growth of the socket and/or shaft cells. Another possibility is that the side of the socket cell crescent marked by Pk at 24 hr is anchored, while the other side grows to wrap around the shaft, inducing junctional rearrangements and propelling the rotation of the apical portion of the shaft relative to the socket cell. Apical rotation could then cause twisting of the more basal portions of the socket cell, and could in turn direct the shaft to the corresponding side of the socket cell.

The precise location at which the PCP signal is required to determine rotation direction is unclear. Because we observe asymmetric core complexes at margin-socket cell junctions, but very little at shaft-margin or shaft-socket junctions, we hypothesize that interaction between the socket and surrounding margin cells is the essential determinant of rotation. PCP proteins at these junctions could control junctional dynamics, as is known to occur in other systems (***Huebner and Wallingford, 2018***). This will require further investigation.

## Role of core PCP signaling in socket-shaft rotation

Core PCP signaling participates in regulating rotation, as the magnitude of rotation is substantially impaired in the core mutants *fz*, *dsh*, *vang* and *fmi*. Nonetheless, we note that a small amount of clockwise rotation still occurs in these mutants. We hypothesize that tissue scale mechanical forces may drive this rotation, though we do not rule out the possibility that some other signaling activity may also be involved. Compared to other core proteins, the impact of removing Pk$^{pk}$ on the magnitude of rotation is subtle. The clockwise rotation in *pk$^{pk-sple}$* mutants is only slightly less than in wildtype. This result is reminiscent of Pk$^{pk}$ function in polarizing wing hairs: polarity in *pk$^{pk}$* mutants is strongly perturbed due to the presence of Pk$^{sple}$, while hair polarity in *pk$^{pk-sple}$* mutants is only weakly perturbed (*Gubb et al., 1999*). These findings are consistent with previous proposals that the core PCP mechanism retains a residual capacity to propagate some asymmetry in the absence of Pk (*Strutt and Strutt, 2007*; *Lawrence et al., 2004*; *Adler et al., 2000*).

The core PCP signal, in addition to executing directed rotation in response to Pk$^{pk}$ or Pk$^{sple}$, coordinates polarity between neighboring bristles. Local correlation between rotation angles is strong when core signaling is intact, even in the absence of the Ft/Ds signal, but is weak when core signaling is disrupted. This is analogous to the proposed mechanism for locally coordinating polarity between adjacent wing hairs. It is important to note that the local polarity signal must pass through intervening margin cells to signal from bristle to bristle.

## Spatiotemporal dynamics and selection of Pk$^{pk}$ versus Pk$^{sple}$ for polarity determination

Our data suggest that whether the Pk$^{pk}$ or Pk$^{sple}$ isoform dominates to control the direction of PCP signaling depends not only on the relative amounts of each isoform, but also on the dynamics of expression and its effect on competition for participation in the core complex. During rotation of AWM bristle shaft-socket pairs, both Pk$^{pk}$ and Pk$^{sple}$ isoforms are detected at the apical junction of the socket with an inverse temporal relationship; high expression of Pk$^{pk}$ decreases during rotation, while the initially undetectable level of Pk$^{sple}$ protein increases. In these conditions, the system is controlled by Pk$^{pk}$, and both Pk$^{pk}$ and Pk$^{sple}$ localize proximally, thus orienting the core complex in its wildtype configuration. We hypothesize that Pk$^{sple}$ is recruited by Pk$^{pk}$ through their known ability to interact heterotypically (*Ayukawa et al., 2014*; *Ambegaonkar and Irvine, 2015*). This ability of Pk$^{pk}$ and Pk$^{sple}$ to colocalize has not been previously observed in wildtype conditions. Notably, however, in the wing, ectopic Pk$^{sple}$ localization follows the expected position of Pk$^{pk}$ when Ds and Dachs cues are removed, though each were not independently visualized (*Ambegaonkar and Irvine, 2015*). Conversely, Pk$^{sple}$ overexpression was seen to recruit Pk$^{pk}$ to the distal side of wing cells (*Ayukawa et al., 2014*), reversing hair polarity as it does bristle polarity. We suggest that the temporal expression pattern in the AWM allows the system to initiate polarization under Pk$^{pk}$ control, and that the gradually accumulating Pk$^{sple}$ colocalizes with Pk$^{pk}$ rather than outcompeting established proximal localization. Because bristles in *pk$^{sple}$* mutant wings fully polarize, the proximal Pk$^{sple}$ is inconsequential for normal bristle polarization.

In contrast, overexpression of Pk$^{sple}$, producing early and sustained high level expression, enables it to outcompete endogenous Pk$^{pk}$ and reverse polarity by driving localization to the distal side through its interaction with Ds and Dachs, likely recruiting Pk$^{pk}$ along with it. Similarly, in *pk$^{pk}$* mutants, endogenous Pk$^{sple}$, free from recruitment to the proximal side, localizes distally and reverses polarity. We infer that during the critical period for determining bristle rotation direction in wildtype, Pk$^{sple}$ does not reach a sufficient level to outcompete Pk$^{pk}$ and reverse the rotation.

That *pk$^{pk}$* mutation only reverses polarity of a region of distal bristles, whereas Pk$^{sple}$ overexpression can reverse polarity of most or all AWM bristles, indicates that endogenous Pk$^{sple}$ is only poised to act in a limited region of the margin. This may reflect subtle differences in the timing of its expression increase across the margin. Alternatively, it may reflect differences in the strength of the Ft/Ds/Fj signal across the margin. Our analyses do not have sufficient resolution to distinguish these possibilities.

Dynamic isoform expression appears to have important consequences for other aspects of wing development. Hair polarity is determined by Pk$^{pk}$ (at around 32 hr apf), but it can be inferred that some Pk$^{sple}$ is already present, as is evident from the difference between the hair polarity patterns of *pk$^{pk}$* and *pk$^{pk-sple}$* mutants (*Gubb et al., 1999*), and as confirmed by our expression analyses. We

suggest that hair polarity does not fully reverse in $pk^{pk}$ mutants either because levels of Pk$^{sple}$ are not yet high enough or because expression is primarily in veins and at the AWM at the time hair polarity is fixed. In contrast, the polarity of ridges, established later in wing development [(*Merkel et al., 2014*); (*Doyle et al., 2008*) notwithstanding], depends on Pk$^{sple}$. We propose that by the time ridge polarity is determined, the amount of Pk$^{sple}$ has increased and the amount of Pk$^{pk}$ has decreased sufficiently to allow Pk$^{sple}$ to exert control of ridge polarization. Though likely unimportant for normal development, the somewhat earlier expression of Pk$^{sple}$ in veins relative to the intervein regions may contribute to polarity discontinuities observed in $pk^{pk}$ mutant wings, especially around L3 (*Gubb and García-Bellido, 1982*; *Hogan et al., 2011*; *Merkel et al., 2014*). Ft-Ds polarity appears to also be distorted around veins (*Merkel et al., 2014*). Pk$^{pk}$ and Pk$^{sple}$ expression dynamics are likely at play in determining the PCP response in other tissues as well.

## The ft/Ds/Fj global signal orients both Pk$^{pk}$-dependent and Pk$^{sple}$-dependent core PCP signaling in the wing

The idea that Ds controls the direction of core PCP signaling was first proposed by Adler based on wing hair polarity phenotypes (*Adler et al., 1998*). We subsequently studied the Ft/Ds/Fj system and similarly concluded that it directs core PCP protein localization in the wing (*Ma et al., 2003*), a Pk$^{pk}$-dependent process, and polarization of ommatidia in the eye (*Yang et al., 2002*), a Pk$^{sple}$-dependent process. We proposed that coupling in wing hair polarization is necessarily weak (*Ma et al., 2003*), and the more recently proposed model in which Ft/Ds/Fj orient microtubules to orient directional trafficking of Fz, Dsh and Fmi-containing vesicles (*Matis et al., 2014*; *Olofsson and Axelrod, 2014*; *Shimada et al., 2006*; *Harumoto et al., 2010*) is consistent with a weak coupling mechanism in Pk$^{pk}$-dependent processes. The finding of direct binding of Pk$^{sple}$ to Ds and Dachs (*Ayukawa et al., 2014*; *Ambegaonkar and Irvine, 2015*) suggests a model for more direct and potentially stronger coupling of Pk$^{sple}$-dependent processes to the Ft/Ds/Fj system. The idea of coupling in Pk-dependent signaling has been controversial, and based largely on correlation, subsequent studies have led to the argument that Pk$^{sple}$-dependent core signaling is coupled, but Pk$^{pk}$-dependent signaling is uncoupled from the the Ft/Ds/Fj system (*Merkel et al., 2014*; *Ambegaonkar and Irvine, 2015*). Yet others have suggested that the Ft/Ds/Fj and core PCP systems always function in parallel rather than being coupled (*Lawrence et al., 2007*; *Casal et al., 2006*). Here, we report strong evidence that Pk$^{pk}$-dependent core PCP signaling is responsive to the Ds signal, at least in polarizing wing hairs. We propose that the same is the case in polarizing bristles that are controlled by essentially similar responses to Pk isoforms and to the Ft/Ds/Fj system.

In bristles, we directly observe the requirement for Ds to distally localize Pk$^{sple}$ when Pk$^{pk}$ is absent, confirming Pk$^{sple}$ coupling. The evidence that Pk$^{pk}$-dependent core signaling is coupled to the upstream Ft/Ds/Fj signal is less apparent. In wildtype, correct bristle polarization requires Pk$^{pk}$ to prevent reversal by recruiting Pk$^{sple}$ to the proximal side, though as noted above, proximal Pk$^{sple}$ plays no essential role. But absent the need to antagonize Pk$^{sple}$ coupling, is there evidence that core signaling in the presence of just Pk$^{pk}$ ($pk^{sple}$ mutant) is coupled to Ft/Ds/Fj? When the Ft/Ds/Fj system is intact, Pk$^{pk}$ localizes proximally, but without Ds or Ft, Pk$^{pk}$ localizes proximally and distally in random domains, driving domains of correct and reversed rotation analogous to the random but locally correlated domains of hair polarity in *ft* or *ds* mutant wing tissue (*Adler et al., 1998*; *Ma et al., 2003*). The same random domains of bristle polarity are seen when only Pk$^{pk}$ is available (*ds* knockdown in a $pk^{sple}$ mutant). This result demonstrates that the Ft/Ds/Fj system is required for correct polarization of the core PCP system while solely under Pk$^{pk}$ control, though it cannot distinguish a permissive from an instructive role. An instructive role is, however, concordant with its instructive role in directing Pk$^{pk}$-dependent wing hair polarity.

The proposal that Pk$^{pk}$-dependent core signaling is coupled to and responds to the Ft/Ds/Fj signal in bristle polarization might at first appear to conflict with the observation that properly oriented rotation proceeds to a significant extent in the absence of both Pk$^{pk}$ and Pk$^{sple}$. This is explained by pointing out that our model for coupling invokes Ft/Ds/Fj directed microtubule-based transport of Fz and Dsh, but that the involvement of Pk$^{pk}$ is indirect (*Matis et al., 2014*; *Olofsson and Axelrod, 2014*; *Shimada et al., 2006*; *Harumoto et al., 2010*). As have others, we propose that Pk$^{pk}$ functions to amplify the asymmetry introduced by this transport, but that some asymmetry, and communication of polarity information between cells, can still occur in its absence (*Strutt and Strutt, 2007*; *Lawrence et al., 2004*; *Adler et al., 2000*). In other words, Ft/Ds/Fj coupling to Pk$^{pk}$-dependent

core PCP signaling is not directed by Pk$^{pk}$, but rather, is permitted to occur because the Pk$^{sple}$-dependent mechanism is not operating to override it. Because core module function is required, this activity does not result from Pk isoform action influencing Ft/Ds/Fj output independent of core signaling, as has been recently suggested in another context (*Casal et al., 2018*). It is important to caution that the model for the relationship between Ft/Ds/Fj and core signaling presented here does not necessarily extend to their relationship in other tissues where their interactions may well be different, and that experiments done in other tissues may not be directly relevant to wing hair and AWM bristles.

The results presented here indicate that mapping the spatiotemporal dynamics of Pk isoform expression is essential to understanding how various developmental events can be differentially coupled to upstream global directional signals in a given tissue.

## A hypothesis for translating PCP into organ rotation

Chirality, or left-right laterality, is a key feature of many organs in invertebrates and vertebrates. In *Drosophila*, rotation of the gut and of the male genitalia occurs in a defined direction to produce such laterality. In vertebrates, rotation of the gut and heart tube also leads to left-right asymmetry in these organs. In many cases, PCP has been implicated in control of this lateralization (*Blum and Ott, 2018*). In the *Drosophila* hindgut, both core PCP and the Ft/Ds system play essential roles in directing normal dextral rotation (*González-Morales et al., 2015*). Though the forces that drive these organ rotations are not well understood, left-right asymmetries in actomyosin distribution, cell shape, and localization of other cellular structures, together with PCP dependence (*Harris, 2018*; *Blum and Ott, 2018*), indicate that chirality at the cellular level is an important determinant of rotational direction. Indeed, chirality of isolated cells from looping chick heart has been directly demonstrated (*Ray et al., 2018*). We therefore propose that regulation of chiral shaft-socket cell pair rotation may share much in common with the mechanisms that determine larger organ laterality, and its investigation could therefore yield insights that will enlighten understanding of organ rotation and laterality.

## Materials and methods

### Key resources table

| Reagent type (species) or resource | Designation | Source or reference | Identifiers | Additional information |
|---|---|---|---|---|
| Genetic reagent (*Drosophila melanogaster*) | pk$^{pk-sple13}$ | *Gubb et al., 1999*, PMID: 10485852 | BDSC:41790; FLYB:FBal0060943; RRID:BDSC_41790 | FlyBase symbol: pk$^{pk-sple-13}$ |
| Genetic reagent (*Drosophila melanogaster*) | pk$^{pk-sple14}$ | *Gubb et al., 1999*, PMID: 10485852 | FLYB:FBal0035401 | FlyBase symbol: pk$^{pk-sple-14}$ |
| Genetic reagent (*Drosophila melanogaster*) | pk$^{pk30}$ | *Gubb et al., 1999*, PMID: 10485852 | BDSC:44229; FLYB:FBal0101223; RRID:BDSC_44229 | FlyBase symbol: pk$^{30}$ |
| Genetic reagent (*Drosophila melanogaster*) | pk$^{sple1}$ | *Gubb et al., 1999*, PMID: 10485852 | BDSC:422; FLYB:FBal0016024; RRID:BDSC_422 | FlyBase symbol: pk$^{sple-1}$ |
| Genetic reagent (*Drosophila melanogaster*) | vang$^{A3}$ | *Taylor et al., 1998*, PMID: 9725839 | FLYB:FBal0093183 | FlyBase symbol: Vang$^{A3}$ |
| Genetic reagent (*Drosophila melanogaster*) | vang$^{stbm6}$ | *Wolff and Rubin, 1998*, PMID: 9463361 | BDSC:6918; FLYB:FBal0062424; RRID:BDSC_6918 | FlyBase symbol: Vang$^{stbm-6}$ |
| Genetic reagent (*Drosophila melanogaster*) | fz$^{R52}$ | *Krasnow and Adler, 1994*, PMID: 7924994 | FLYB:FBal0004939 | FlyBase symbol: fz$^{23}$ |
| Genetic reagent (*Drosophila melanogaster*) | dsh$^{1}$ | Bloomington *Drosophila* Stock Center | BDSC:5298; FLYB:FBal0003138; RRID:BDSC_5298 | FlyBase symbol: dsh$^{1}$ |

*Continued on next page*

*Continued*

| Reagent type (species) or resource | Designation | Source or reference | Identifiers | Additional information |
|---|---|---|---|---|
| Genetic reagent (*Drosophila melanogaster*) | *UAS-pk^sple* | Bloomington *Drosophila* Stock Center | BDSC:41780; FLYB:FBti0148928; RRID:BDSC_41780 | FlyBase symbol: *P{UAS-sple^+}3* |
| Genetic reagent (*Drosophila melanogaster*) | *UAS-pk^RNAi* | Vienna *Drosophila* Resource Center | VDRC:v101480; FLYB:FBst0473353; RRID:FlyBase_FBst0473353 | FlyBase symbol: *P{KK109294}VIE-260B* |
| Genetic reagent (*Drosophila melanogaster*) | *UAS-fmi^RNAi* | Bloomington *Drosophila* Stock Center | BDSC:26022; FLYB:FBti0114752; RRID:BDSC_26022 | Flybase symbol: *P{TRiP.JF02047}attP2* |
| Genetic reagent (*Drosophila melanogaster*) | *UAS-fz^RNAi* | Bloomington *Drosophila* Stock Center | BDSC:34321; FLYB:FBti0140932; RRID:BDSC_34321 | Flybase symbol: *P{TRiP.HMS01308}attP2* |
| Genetic reagent (*Drosophila melanogaster*) | *UAS-vang^RNAi* | Bloomington *Drosophila* Stock Center | BDSC:34354; FLYB:FBti0140967; RRID:BDSC_34354 | Flybase symbol: *P{TRiP.HMS01343}attP2* |
| Genetic reagent (*Drosophila melanogaster*) | *UAS-ds^RNAi* | Bloomington *Drosophila* Stock Center | BDSC:32964; FLYB:FBti0140473; RRID:BDSC_32964 | Flybase symbol: *P{TRiP.HMS00759}attP2* |
| Genetic reagent (*Drosophila melanogaster*) | *UAS-ds* | *Matakatsu and Blair, 2004*, PMID: 15240556 | FLYB:FBtp0019964 | Flybase symbol: *P{UAS-ds.T}* |
| Genetic reagent (*Drosophila melanogaster*) | *dll-GAL4* | Bloomington *Drosophila* Stock Center | BDSC:3038; FLYB:FBti0002783; RRID:BDSC_3038 | Flybase symbol: *P{GawB}Dll^md23* |
| Genetic reagent (*Drosophila melanogaster*) | *MS1096-GAL4* | Bloomington *Drosophila* Stock Center | BDSC:8860; FLYB:FBti0002374; RRID:BDSC_8860 | Flybase symbol: *P{GawB}Bx^MS1096* |
| Genetic reagent (*Drosophila melanogaster*) | *armP-fz::EGFP* | *Strutt, 2001*, PMID: 11239465 | FLYB:FBtp0014592 | Flybase symbol: *P{arm-fz.GFP}* |
| Genetic reagent (*Drosophila melanogaster*) | *actP-vang::EYFP* | *Strutt, 2002*, PMID: 12137731 | FLYB:FBtp0015854 | Flybase symbol: *P{Act5C(-FRT)stbm-EYFP}* |
| Genetic reagent (*Drosophila melanogaster*) | *actP > CD2>vang::EYFP* | *Strutt, 2002*, PMID: 12137731 | FLYB:FBtp0084387 | Flybase symbol: *P{Act5C(FRT.polyA)stbm-EYFP}* |
| Genetic reagent (*Drosophila melanogaster*) | *ci-GAL4* | *Croker et al., 2006*, PMID: 16413529 | FLYB:FBtp0057188 | Flybase symbol: *P{ci-GAL4.U}* |
| Genetic reagent (*Drosophila melanogaster*) | *UAS-mCherry* | Bloomington *Drosophila* Stock Center | BDSC:38424; FLYB:FBti0147460; RRID:BDSC_38424 | Flybase symbol: *P{UAS-mCherry.NLS}3* |
| Genetic reagent (*Drosophila melanogaster*) | *actP > CD2>Gal4* | Bloomington *Drosophila* Stock Center | BDSC:30558; FLYB:FBti0012408; RRID:BDSC_30558 | Flybase symbol: *P{GAL4-Act5C(FRT.CD2).P}S* |
| Genetic reagent (*Drosophila melanogaster*) | *UAS-RFP* | Bloomington *Drosophila* Stock Center | BDSC:30558; FLYB:FBti0129814; RRID:BDSC_30558 | Flybase symbol: *P{UAS-RFP.W}3* |
| Antibody | goat polyclonal anti-Su(H) | Santa Cruz | Santa Cruz:sc-15183 RRID:AB_672840 | 1/200 (immunolabelling) |
| Antibody | Mouse monoclonal anti-V5 | Thermo-Fisher | Thermo_Fisher:R960-25, RRID:AB_2556564 | 1/200 (immunolabelling) 1/1000 (Western blotting) |
| Antibody | Guinea pig polyclonal anti-Pk[C] | *Olofsson et al., 2014*, PMID: 25005476 | N/A | 1/800 (immunolabelling) 1/1000 (Western blotting) |
| Antibody | Rat monoclonal anti-dEcad | DSHB | RRID:AB_528120 | 1/200 (immunolabelling) |
| Antibody | Mouse monoclonal anti-γ-Tubulin | Sigma-Aldrich | Sigma-Aldrich: T6557 RRID:AB_477584 | 1/1000 (Western blotting) |
| Recombinant DNA reagent | pCFD4 | Addgene | RRID:Addgene_49411 | CRISPR gRNA backbone |

*Continued on next page*

*Continued*

| Reagent type (species) or resource | Designation | Source or reference | Identifiers | Additional information |
|---|---|---|---|---|
| Recombinant DNA reagent | pDsRedattp | Addgene | RRID:Addgene_51019 | Donor recombinant DNA backbone |
| Recombinant DNA reagent | pCR-Blunt-II-TOPO | Thermo-Fisher | RRID:Addgene_29705 | Backbone for sub-cloning |
| Sequence-based reagent | *pk^pk* gRNA 1 | This paper | gRNA sequence in PCR primers | ATGGCTCAGGCCCGATCTAG |
| Sequence-based reagent | *pk^pk* gRNA 2 | This paper | gRNA sequence in PCR primers | GTGGATCAACCCCTGGAAAC |
| Sequence-based reagent | *pk^sple* gRNA 1 | This paper | gRNA sequence in PCR primers | CTCGTAAATTTAGCTTCGAG |
| Sequence-based reagent | *pk^sple* gRNA 2 | This paper | gRNA sequence in PCR primers | AGATGCAATTTGGCCGCCCT |

## *Drosophila* genetics

*Drosophila melanogaster* flies were grown on standard cornmeal/agar/molasses media at 25°C. FLP-on (using the *actP >CD2>GAL4* construct for trans-gene expression) and FLP/FRT mitotic clones were generated by incubating third-instar larvae at 37°C for 1 hr. 36 to 48 hr later, white prepupae were collected and aged to the desired developmental time point prior to dissection and fixation.

*Drosophila* mutant alleles and transgenic stocks are described in the Key resources table and detailed chromosomes and genotypes are provided below. *pk^{pk-sple13}* (FBst0044230), *pk^{pk-sple14}* (**Gubb, 1993**), *pk^{pk30}* (FBst0044229), *pk^{sple1}* (FBst0000422), *vang^{A3}* (**Taylor et al., 1998**), *vang^{stbm6}* (FBst0006918), *fz^{R52}* (**Krasnow and Adler, 1994**), *dsh^1* (FBst0005298), *UAS-pk^sple* (FBst0041780), *UAS-pk^{RNAi}* (VDRC ID: 101480), *UAS-fmi^{RNAi}* (FBst0026022), *UAS-fz^{RNAi}* (FBst0034321), *UAS-vang^{RNAi}* (FBst0034354), *UAS-ds^{RNAi}* (FBst0032964), *UAS-ds* (**Matakatsu and Blair, 2004**), *dll-Gal4* (FBst0030558), *MS1096-Gal4* (FBst0008860), *armP-fz::EGFP* (**Strutt, 2001**), *actP-vang::EYFP* (**Strutt, 2002**), *actP >CD2>vang::EYFP* (**Strutt, 2002**), *ci-Gal4* (**Croker et al., 2006**), *UAS-mCherry* (FBst0038424), *actP >CD2>Gal4 UAS-RFP* (FBst0030558).

## Genotypes of experimental models

### *Figure 1*

(A, C) *w^{1118}*
(B, D) *y w hsflp/+;; actP >CD2>GAL4 UAS-RFP/+*

### *Figure 2*

(A) *pk^{pk30}/pk^{pk30}*
(B) *y w hsflp/+; pk^{pk30}/pk^{pk30}; actP >CD2>GAL4 UAS-RFP/+*

### *Figure 3*

(A) *y w hsflp/+;; actP >CD2>GAL4 UAS-RFP/+*
(D) *w^{1118}*
(E) *pk^{pk30}/pk^{pk30}*

### *Figure 4*

(A, C, E) *V5::3Xmyc::APEX2::pk^{pk}/V5::3Xmyc::APEX2::pk^{pk}*
(B, D, F) *V5::3Xmyc::APEX2::pk^{sple}/V5::3Xmyc::APEX2::pk^{sple}*
(G) *pk^{pk30}/pk^{pk30}*
(H) *y w hsflp/+; V5::3Xmyc::APEX2::pk^{pk}/UAS-pk^{RNAi}; actP >CD2>GAL4 UAS-RFP/+*
(I) *y w hsflp/+; V5::3Xmyc::APEX2::pk^{sple}/UAS-pk^{RNAi}; actP >CD2>GAL4 UAS-RFP/+*

## Figure 5

(A) $w^{1118}$
(B) $pk^{pk30}/pk^{pk30}$
(C) $MS1096\text{-}GAL4/+; pk^{pk30}/pk^{pk30};UAS\text{-}fz^{RNAi}/+$
(D) $MS1096\text{-}GAL4/+; pk^{pk30}/pk^{pk30};UAS\text{-}vang^{RNAi}/+$
(E) $MS1096\text{-}GAL4/+;;UAS\text{-}fmi^{RNAi}/+$
(F) $fz^{R52}/fz^{R52}$
(G) $vang^{stbm6}/vang^{stbm6}$
(H) $dsh^1/dsh^1$
(I) $armP\text{-}fz::EGFP/armP\text{-}fz::EGFP$
(J) $actP\text{-}vang::EYFP/actP\text{-}vang::EYFP$
(K) $y\ w\ hsflp/+; FRT42D\ armP\text{-}fz::EGFP/FRT42D$
(L) $y\ w\ hsflp/+; FRT42D\ actP\text{-}vang::EYFP/FRT42D$

## Figure 6

y w hsflp/+; armP-fz::EGFP/+; actP >CD2>GAL4 UAS-RFP/UAS-fz$^{RNAi}$

## Figure 7

(A) $w^{1118}$
(B) $pk^{pk30}/pk^{pk30}$
(C) $MS1096\text{-}GAL4/+; V5::3Xmyc::APEX2::pk^{sple}/+; UAS\text{-}ds^{RNAi}/+$
(D) $MS1096\text{-}GAL4/+; pk^{pk30}/pk^{pk30}; UAS\text{-}ds^{RNAi}/+$
(E) $MS1096\text{-}GAL4/+; pk^{sple1}/pk^{sple1}; UAS\text{-}ds^{RNAi}/+$
(F) $MS1096\text{-}GAL4/+; pk^{pk\text{-}sple13}/pk^{pk\text{-}sple13}; UAS\text{-}ds^{RNAi}/+$
(G) $w^{1118}$ and $dll\text{-}GAL4/+; UAS\text{-}ds/UAS\text{-}ds$
(H) $fz^{R52}/fz^{R52}$ and $dll\text{-}GAL4/+; fz^{R52}\ UAS\text{-}ds/fz^{R52}\ UAS\text{-}ds$
(I) $pk^{pk30}/pk^{pk30}$ and $pk^{pk30}\ dll\text{-}GAL4/pk^{pk30}; UAS\text{-}ds/UAS\text{-}ds$
(J) $pk^{pk\text{-}sple13}/pk^{pk\text{-}sple14}$ and $dll\text{-}GAL4\ pk^{pk\text{-}sple13}/pk^{pk\text{-}sple14}; UAS\text{-}ds/UAS\text{-}ds$
(K) $pk^{sple1}/pk^{sple1}$ and $pk^{sple1}\ dll\text{-}GAL4/pk^{sple1}; UAS\text{-}ds/UAS\text{-}ds$

## Figure 1—figure supplement 1

(B) $w^{1118}$
(C) $pk^{pk30}/pk^{pk30}$

## Figure 2—figure supplement 1

(A2) $w^{1118}$ and $pk^{pk30}/pk^{pk30}$
(B) $actP\text{-}vang::EYFP/+$

## Figure 3—figure supplement 1

(A) $w^{1118}$
(B) $pk^{pk30}/pk^{pk30}$
(C) $pk^{sple1}/pk^{sple1}$
(D) $pk^{pk\text{-}sple13}/pk^{pk\text{-}sple13}$
(E) $MS1096\text{-}GAL4/+;;UAS\text{-}pk^{sple}/+$

## Figure 4—figure supplement 1

(A) $V5::3Xmyc::APEX2::pk^{pk}/V5::3Xmyc::APEX2::pk^{pk}$,
$V5::3Xmyc::APEX2::pk^{sple}/V5::3Xmyc::APEX2::pk^{sple}$
(B)Lane 1: $pk^{pk\text{-}sple13}/pk^{pk\text{-}sple13}$, other lanes: $w^{1118}$
(C-H) $V5::3Xmyc::APEX2::pk^{pk}/V5::3Xmyc::APEX2::pk^{pk}; ci\text{-}GAL4\ UAS\text{-}mCherry/+$
(K)$V5::3Xmyc::APEX2::pk^{pk}/V5::3Xmyc::APEX2::pk^{pk}$

(I-J, L) *V5::3Xmyc::APEX2::pk^sple^/V5::3Xmyc::APEX2::pk^sple^*

## Figure 4—figure supplement 2

(A) *V5::3Xmyc::APEX2::pk^pk^/V5::3Xmyc::APEX2::pk^pk^*,
(B) Left: *V5::3Xmyc::APEX2::pk^sple^/V5::3Xmyc::APEX2::pk^sple^; actP-vang::EYFP/+*
Middle and right: *V5::3Xmyc::APEX2::pk^sple^/V5::3Xmyc::APEX2::pk^sple^*
(C) *pk^pk30^/pk^pk30^*
(D) *y w hsflp/+; pk^pk30^/pk^pk30^ UAS-pk^RNAi^; actP >CD2>GAL4 UAS-RFP/+*

## Figure 5—figure supplement 1

*y w hsflp/+; pk^pk30^/pk^pk30^; actP >CD2>vang::EYFP/+*

## Figure 6—figure supplement 1

*y w hsflp/+; actP-vang::EYFP/+; actP >CD2>GAL4 UAS-RFP/UAS-vang^RNAi^*

## Figure 7—figure supplement 1

(A) *MS1096-GAL4/+;; UAS-ds^RNAi^/+*
(B) *MS1096-GAL4/+; pk^pk30^/pk^pk30^; UAS-ds^RNAi^/+*
(C) *MS1096-GAL4/+; pk^sple1^/pk^sple1^; UAS-ds^RNAi^/+*
(D) *MS1096-GAL4/+; pk^pk-sple13^/pk^pk-sple13^; UAS-ds^RNAi^/+*
(E) *dll-GAL4/+; UAS-ds/UAS-ds*
(F–I) *dll-GAL4/+; UAS-ds/UAS-mCherry*

## Immunohistochemistry

Pupal wings were dissected at indicated developmental time points after puparium formation (apf). Pupae were removed from their pupal cases and fixed for 60–90 min in 4% paraformaldehyde in PBS at 4°C. Wings were then dissected and extracted from the cuticle, and were washed two times in PBST (PBS with 0.1% Triton X-100). After blocking for 1 hr in 5% Bovine serum Albumin in PBST at 4°C, wings were incubated with primary antibodies overnight at 4°C in the blocking solution. Incubations with secondary antibodies were done for 90 min at room temperature in PBST. Washes in PBST were performed three times after primary and secondary antibody incubation, and incubations in phalloidin (1:200 dilution) in PBST were done for 15 min followed by wash at room temperature before mounting if required. Stained wings were mounted in 15 μl Vectashield mounting medium (Vector Laboratories). Primary antibodies were as follows: goat polyclonal anti-Su(H) (1:200 dilution, Santa Cruz, sc-15183), mouse monoclonal anti-V5 (1:200 dilution, Thermo-fisher, R960-25), guinea pig polyclonal anti-Pk[C] (1:800 dilution, *Olofsson et al., 2014*), rat monoclonal anti-dEcad (1:200 dilution, DSHB). Secondary antibodies from Thermo Fisher Scientific were as follows: 488-donkey anti-mouse, 488-goat anti-guinea pig, 546-donkey anti-goat, 633-goat anti-guinea pig, 633-goat anti-rat, 647-donkey anti-mouse. Alexa 635 and Alexa 350 conjugated phalloidin were from Thermo Fisher Scientific.

## Imaging and quantification

Adult wings were dissected and washed with 70% EtOH and mounted in DPX (Sigma) solution. All adult wings were imaged on a Nikon Eclipse E1000M equipped with a Spot Flex camera (Model 15.2 64 MP). All immunofluorescence images were taken with a Leica TCS SP8 AOBS confocal microscope and processed with LAS X (Leica) and Adobe Photoshop. For three dimensional wing margin images, 50 to 100 z-stacks, each with 0.2 μm thickness, were collected and combined using 3D reconstitution software (Leica). Scale bars are not provided for three dimensional images due to errors introduced by perspective, but approximate scale can be inferred from related two- dimensional images. To measure the apical rotation angles of socket cells, a horizontal line linking centers of circles around apical surfaces of socket cells was drawn, and perpendicular lines intersecting the

center of each socket cell apex were drawn (black lines in *Figure 3C*). Vectors from each socket cell center passing through the center of the apical opening of the socket circles (blue vectors in *Figure 3C*) were drawn and angles between the vertical lines and the vectors were measured with Image J software. Statistical analysis was performed and rose plots generated using Oriana four software. Comparisons were made using Student's t-test and p values are reported. Summary statistics are provided in *Table 2*. For qualitative results such as expression patterns, a minimum of 20 biological replicates from at least two independent experiments were examined and representative images are shown.

## CRISPR/Cas9 homology directed recombination for tagging V5 sequence to $pk^{pk}$ and $pk^{sple}$ genomic locus

### Construction of gRNA containing plasmids

$pk^{pk}$ gRNA1 (5'-ATGGCTCAGGCCCGATCTAG-3') and $pk^{pk}$ gRNA2 (5'- GTGGATCAACCCC TGGAAAC-3') were assembled into *pCFD4* plasmid (Addgene, 49411) digested by the BbsI restriction enzyme using Gibson Assembly (NEB) to express two gRNAs under the U6 promoter. The same procedure was carried out to assemble two $pk^{sple}$ gRNAs, $pk^{sple}$ gRNA1 (5'-CTCGTAAATTTAGC TTCGAG-3') and $pk^{sple}$ gRNA2 (5'- AGATGCAATTTGGCCGCCCT-3'), into *pCFD4*. Stable transgenic flies expressing two $pk^{pk}$ gRNAs or two $pk^{sple}$ gRNAs were generated by BestGene using the PhiC31 standard injection method.

## Construction of donor plasmids containing two homology arms and V5::3Xmyc::APEX2 tag sequences

To generate the donor plasmid with homology arms of the $pk^{pk}$ genomic sequence and the *V5::3Xmyc::APEX2* tag sequence, a 1.5 kb 3' homology arm (HR2) flanking the $pk^{pk}$ gRNA2 cleavage site was amplified and assembled into the *pDsRed-attp* (Addgene, 51019) plasmid cut with SapI to make the *pDsREd-attP-pk^{pk}HR2* plasmid.

To generate the donor plasmid for tagging $pk^{pk}$, three DNA fragments including 5' 1.5 kb homology arm (HR1; containing a 1.2 kb homology arm flanking the $pk^{pk}$ gRNA1 cleavage site and a 5' 0.3 kb sequence of the start codon), *V5::3Xmyc::APEX2* tag with a linker sequence, and the fragment starting from the start codon of $pk^{pk}$ to the cleavage site targeted by the $pk^{pk}$ gRNA2, were assembled into the *pDsREd-attP-pk^{pk}HR2*. To prevent the donor sequence from being cleaved by Cas9, a point mutation was introduced in the PAM sequence of the HR1 using the NEB point-mutagenesis kit after sub-cloning the fragment into the *pCR-Blunt-II-TOPO* vector (Thermo-Fisher, K280002). The HR1 fragment bearing the mutant PAM sequence was then amplified for the assembly process. All three fragments were assembled into the *pDsREd-attP-pk^{pk}HR2* plasmid cut with EcoRI and NheI.

To generate the donor plasmid for tagging $pk^{sple}$, similar strategies were applied. Briefly, 1.2 kb 5' homology arm containing the mutant PAM sequence, the *V5::3Xmyc::APEX2* tag with a linker sequence, and the fragment from the start codon of $pk^{sple}$ to the cleavage site of $pk^{sple}$ gRNA2 were assembled into the *pDsREd-attP-pk^{sple}HR2* (bearing the 1.25 kb 3' homology arm, HR2) plasmid.

The donor plasmids containing the tag sequence and $pk^{pk}$ homology, or $pk^{sple}$ homology, arms, were sequenced and then injected into the stable transgenic flies expressing two $pk^{pk}$ gRNAs, or $pk^{sple}$ gRNAs, and *nosCas9*, respectively, to generate recombinants. DsRed signal in the fly eyes was monitored for selecting the recombinants by BestGene, and *dsRed* and flanking sequences were removed by the Cre-Lox site-specific recombination method. The resulting modified alleles are referred to in the text as *V5::Pk* and *V5::Sple* for simplicity.

## Western blots

Third-instar larval wing discs and pupal wings at appropriate developmental stages were dissected and lysed in protein loading buffer. Lysates from eight discs or wings were loaded per lane for SDS-PAGE analysis and western blots were performed using standard procedures. Antibodies: Guinea pig polyclonal anti-Pk[C] (1:1000 dilution, the same antibody used for immunostaining), mouse monoclonal anti-V5 (1:2000 dilution, the same antibody used for immunostaining), mouse monoclonal anti-γ-Tubulin (1:1000 dilution, Sigma-Aldrich, T6557). Secondary antibodies were Peroxidase-conjugated goat anti-guinea pig (1:10000) and goat anti-mouse (1:10000) antibodies (both from

Jackson Immuno Research), and detection used SuperSignal West Pico Chemiluminescent Substrate (Thermo-Fisher, 34080)

## Acknowledgements

We thank Helen McNeill, Lucy O'Brien and members of the Axelrod lab for constructive comments.

## Additional information

### Funding

| Funder | Grant reference number | Author |
| --- | --- | --- |
| National Institute of General Medical Sciences | R01 GM097081 | Jeffrey D Axelrod |
| National Institute of General Medical Sciences | R37 GM059823 | Jeffrey D Axelrod |
| National Institute of General Medical Sciences | R35 GM131914 | Jeffrey D Axelrod |

The funders had no role in study design, data collection and interpretation, or the decision to submit the work for publication.

### Author contributions

Bomsoo Cho, Conceptualization, Investigation, Methodology; Song Song, Conceptualization, Data curation, Methodology, Writing - review and editing; Jeffrey D Axelrod, Conceptualization, Supervision, Funding acquisition, Project administration

### Author ORCIDs

Bomsoo Cho (iD) https://orcid.org/0000-0001-8970-9160
Song Song (iD) https://orcid.org/0000-0002-4587-8592
Jeffrey D Axelrod (iD) https://orcid.org/0000-0001-6094-7392

### Ethics

Animal experimentation: This study was performed in strict accordance with the recommendations in the Guide for the Care and Use of Laboratory Animals of the National Institutes of Health. All of the animals were handled according to approved institutional animal care and use committee (IACUC) protocol (#11840) of Stanford University.

### Decision letter and Author response

Decision letter https://doi.org/10.7554/eLife.51456.sa1
Author response https://doi.org/10.7554/eLife.51456.sa2

## Additional files

### Supplementary files

• Transparent reporting form

### Data availability

All data generated or analysed during this study are included in the manuscript and supporting files. Source data files have been provided for all figures containing quantitative data.

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
