## [Decision Letter]

**Acceptance summary:**

Using the mechanosensory bristles of the *Drosophila* wing, Cho et al. describe a new mode of planar polarized morphogenesis. The authors show that bristle orientation arises when the two major cell types that comprise each bristle – the socket and shaft cell – rotate with a defined chirality like a corkscrew and become intertwined. Rotation is dependent on the core PCP pathway and, strikingly, the chirality of rotation depends on which of two Prickle isoforms (Pk or Sple) predominates. By investigating how endogenously-tagged Prickle isoforms respond to changes in the Fat/Dachsous PCP module, they suggest that the two Pk isoforms orient the core PCP module relative to the Fat/Dachsous directional input in opposite ways. Thus, this paper defines a new PCP-dependent morphogenetic process and provides further insights into the intriguing interplay between Prickle isoforms in PCP. Readers broadly interested in cell polarity and morphogenesis will benefit from reading these new contributions to the field.

**Decision letter after peer review:**

Thank you for submitting your article "Prickle isoforms determine handedness of helical morphogenesis" for consideration by *eLife*. Your article has been reviewed by three peer reviewers, including Danelle Devenport as the Reviewing Editor and Reviewer #1, and the evaluation has been overseen by Utpal Banerjee as the Senior Editor.

The reviewers have discussed the reviews with one another and the Reviewing Editor has drafted this decision to help you prepare a revised submission.

Summary:

Cho et al. investigate how mechanosensory bristles of the *Drosophila* wing margin are planar polarized, finding that PCP-dependent rotation of socket and shaft cells, rather than asymmetric positioning of daughter cells, determines bristle orientation. Prickle (Pk) isoform expression appears to determine the chirality of this rotation, and the authors argue that the two Pk isoforms orient the core PCP module relative to the Fat/Dachsous directional input in opposite ways. Overall, the reviewers felt that the documentation of PCP-mediated helical morphogenesis together with further insights into the intriguing interplay between Pk isoforms makes an important contribution to the field. They raised some issues about data presentation, quantification and statistical analysis that should be addressed prior to publication.

The reviewers were also in agreement that the section on Pk^pk^ coupling to the Fat/Ds system was weak and detracted from the rest of the manuscript. The simplest solution would be to remove this section from the revised manuscript so it can be elaborated in a separate study. However, in case the authors do want to address the issues with this section, those comments can be found at the end and separated from the rest of reviewer points.

Reviewer 1:

1) Figure 4H-I. Mosaic knockdown of pk by RNAi is used to distinguish which side of the cell junction Pk and Sple isoforms reside. The authors conclude that both isoforms are proximal. This is an important result as it is different from previously published studies overexpressing Sple and should be shown more clearly. It is very difficult to distinguish the boundaries between expressing and non-expressing cells, especially considering that margin cells may contribute to the signal.

2) Figure 4—figure supplement 1A-B. The loading control levels are highly variable making it difficult to interpret how Pk and Sple isoform levels change over time. Perhaps tubulin is not an appropriate loading control if it levels too change over time. Can the authors plot Pk and Sple levels relative to Tub loading control? Or use V5 fluorescent intensities from confocal images to compare levels?

3) Figure 4. The authors conclude from the localization of Pk and Sple isoforms "that distal localization of Pk^sple^ in Pk^pk^ mutants drives counter-clockwise rotation of shaft-socket pairs." In addition, the title of this section is "Localization of Pk^pk^ and/or Pk^sple^ determines handedness of helical rotation". I think these are overstated conclusions. The localization certainly correlates with the direction of rotation, but this seems insufficient to conclude this drives rotation, especially since elimination of both isoforms has very little effect on rotation.

4) Figure 7D. This experiment shows that reversed Sple localization in Pk^pk^ mutants depends on Ds, and "we interpret this to indicate that the Ds global signal recruits Pk^sple^ to sites of enriched Ds (distal) in the absence of Pk^pk^". What about in the proximal part of the wing where bristle polarity is normal? Is Sple localized proximally, and independent of Ds? How do the authors interpret the distally restricted phenotype of Pk^pk^ mutants? Is the Ds signal stronger distally?

5) Coupling between the Fz-Vang and Ft-Ds pathways by Pk isoforms. "we interpret this to indicate that the Ds global signal recruits Pk^sple^ to sites of enriched Ds (distal) in the absence of Pk^pk^ (Figure 7B; compare with 7D), which drives reversal of shaft-socket orientation and therefore reversal of bristle polarity." Are Fz and Vang localizations flipped in socket cells with reversed polarity in Pk^pk^ mutants? Although this has been shown previously for wing hairs overexpressing Sple, I feel it is important to show here under endogenous Sple levels to definitely show the core PCP module is flipped, not just a downstream component.

Reviewer 2:

1) A "helical structure of defined chirality" may not be appropriate. It seems to be an anisotropic cell rearrangement directed by distal-proximal and dorsal-ventral polarity. Such rearrangement should change the interface between shaft and socket as shown in Figure 3B and C.

2) It is difficult to evaluate the phenotypic analyses without statistical information. What are the percentages of wings showing the representative phenotypes? How may socket cells were scored from how many wings? How many clones were analyzed and what was the percentages showing the described phenotypes? Some of them can be found in source data files. However, it should be also stated in the text.

3) Figure 4H-I', Figure 2—figure supplement 1B, and Figure 4—figure supplement 2 were unclear. These data should be shown more clearly.

Reviewer 3:

1) As a general comment, it might be helpful to readers to explain that in processes such as left-right asymmetry determination (analogous to the chiral morphogenesis being discussed here) there is symmetry breaking on three axes: apical-basal, and two axes in the plane of the tissue. Hence, I presume PCP specifies clockwise or anticlockwise morphogenesis by specifying polarity in one axis of the plane, in concert with a second polarity cue in the orthogonal axis in the plane (which I take to be an AP oriented cue, ultimately set up by the Hedgehog pathway?). Without this second cue, the PCP cue alone wouldn't be sufficient to specify chirality?

2) Introduction paragraph three: I think consistent nomenclature for proteins in flies is capitalized Roman so should be "Prickle" and "Spiny-legs" (and probably "Pk^Pk^" and "Pk^Sple^")? (Also some genotypes change to all caps italics on figure panels e.g. compare Figure 5E-F to 5G-H).

3) Results paragraph one: typo "Supressor"

4) Figure 4—figure supplement 1A is mislabeled? The blue asterisks ("unidentified") and the purple arrow ("Pk") should be swapped? This would then be consistent with panel B?

5) Subsection “Pk^pk^ and Pk^sple^ differentially interpret directional signals from Ds” imply an "indirect" mechanism for Pk^Pk^ coupling to Ft-Ds, but the Introduction (I think more in line with previous work, including from the authors) suggest an indirect mechanism for core pathway coupling to Ft-Ds via Fz/Dsh/Fmi transport with (presumably) Pk^Pk^ being required for magnitude of core pathway activity but *not* for orientation? This is a complex area, but I think it would be easier for readers if the authors stuck to one view. At least based on the data so far, we don't see any evidence that Pk^Pk^ couples, only that it enhances core pathway activity?

6) Paragraph three of subsection “Pk^pk^ and Pk^sple^ differentially interpret directional signals from Ds”: the authors deduce that in the absence of Pk^Pk^, Pk^Sple^ is recruited to distal cell edges by distally localised Ds, but I find this confusing (at least without further clarification) as Merkel et al. seemed to show pretty clearly that – at least in the wing blade – Ds is oriented primarily on the AP axis from 16h APF onwards, and trichomes orient primarily on the AP axis in *pk^pk^* as well (e.g. Merkel et al. and Gubb et al.). So I would expect Sple to be recruited largely to AP cell edges (maybe with a slight distal bias?) not to distal cell edges. But maybe something happens on the edge of the wing blade to tilt the Ds polarity axis more PD? What do the authors think?

7) Paragraph four subsection “Pk^pk^ and Pk^sple^ differentially interpret directional signals from Ds”: this appears to be a key sentence, as here the authors appear to argue that ds knockdown affects a Pk^Pk^-mediated signal for bristle polarity, however I completely failed to follow the logic that indicates Pk^Pk^ is providing a polarity cue rather than contributing to polarity magnitude (see paragraph two of subsection “Pk^pk^ versus Pk^sple^ isoform expression determines the direction of rotation” and Table 3) and suppressing effects of "ectopic" Sple.

– The key piece of data is (I think) that ds- gives reversals of bristle polarity, and (perhaps surprisingly) these are suppressed by loss of Pk and loss of Pk and Sple, but not by loss of Sple alone. I may be missing something important here, but these results seem to say that having Pk activity causes a disruption in polarity in the absence of Ds, but does nothing to support the view that Pk orients polarity?

– Having given it some thought, I think the key to understanding these bristle phenotypes is the original observation of Gubb and Garcia-Bellido, 1982 that "Trichomes and chaetae express the same polarity" i.e. the bristles do what the adjacent trichomes do. In ds we know trichomes reverse in the mid-anterior wing margin (Adler et al., 1998) and this seems to be due to altered core PCP in turn produced by disrupted cell flows, oriented cell divisions etc. (Aigouy et al., Merkel et al.) [other explanations for the swirling phenotype are possible but don't alter the argument I think]. In ds double mutants with the core pathway, trichomes show a *fz/in* pattern (Adler et al., 1998) and point broadly distal on the anterior wing margin, and we see bristles doing the same in *ds pk-^sple^* i.e. the phenotype of core loss-of-function is epistatic to that of *ft-ds* for trichome polarity. In *ds pk^pk^* we also see trichomes pointing distally as do the bristles (e.g. Hogan et al., 2011), presumably because Sple isn't being mislocalized by Ds (as also suggested in the text by the authors here) and we're seeing either a wild-type or *fz/in* loss-of-function polarity phenotype?

Comments related to trichome polarity:

1) The authors also revisit the issue of whether Ft-Ds instruct trichome polarity in Figure 7G-K. However, these results just seem to agree with the view that the core loss-of-function trichome polarity phenotypes are epistatic to loss- or gain-of-function *ft-ds* trichome polarity phenotypes. There is no dispute that *dll>2x-ds* makes trichome polarity swirl (and as they normally point distally, any swirling can be interpreted as "reversal"), but I think it is hard to prove or disprove whether this is direct effect on the core pathway or an indirect effect on the complex process of wing morphogenesis (c.f. work from Eaton lab) that alters core polarity. I don't think it's a surprise that loss of *fz* or *pk^sple^* suppresses this phenotype (epistasis noted by Adler et al., 1998) or that *pk^sple^* does not (this leaves core pathway intact).

The interesting question is why the *dll>2x-ds; pk^pk^* phenotype looks like the *pk^pk^* phenotype. The authors describe this as a "suppression", and seem to conclude that this means Pk^Pk^ couples to Ft-Ds, but I don't follow the leap of logic. I think the accepted view is that in *pk^pk^* the residual population of Pk^Sple^ is oriented by endogenous Ft-Ds generating the familiar *pk^pk^* AP oriented trichome pattern, and it appears that this AP orientation persists in a *dll>2x-ds* background. This persistence is interesting, but seems to be more about how and when Pk^Sple^ couples to Ft-Ds, and doesn't seem to reveal anything about whether Pk^Pk^ can couple to Ft-Ds. Have I missed something?

2) Overall I feel it's a pretty shaky case that Ft-Ds is directly orienting the core in these experiments or that Pk is acting to couple the core to Ft-Ds. Of course if you did want to argue that Ft-Ds orient the core in the relevant region of the wing blade to affect either distal trichome polarity or the triple row bristles, I think you would have to carefully reconcile the following published observations:

i) Ft-Ds polarity seems to be AP while core is PD, so they don't point the same way? (A timing issue maybe?)

ii) flattening Ds and Fj gradients seems to give normal polarity? (Also of bristles?)

iii) *ft dachs* and ft UAS-ft-intra and ft UAS-wts give normal trichome polarity? (Triple row bristles also point distally in these genotypes?)

However, this all seems like a different argument, not pertinent to helical morphogenesis.

3) Nevertheless, in light of the above, I worry about the statement in the Abstract "Pk^pk^ and Pk^sple^, determines right- or left-handed bristle morphogenesis, each by coupling Frizzled/Vang signaling to the Fat/Dachsous PCP directional signal in opposite directions". On one hand I don't follow the argument showing Pk^Pk^ is coupling to Ft-Ds or that Ft-Ds is determining bristle polarity. The fact that bristle polarity is disrupted in ds mutants doesn't necessarily imply Ds is instructive. I agree that in the absence of Pk^Pk^, Pk^Sple^ is probably reversing bristle polarity by coupling to Ds – but this doesn't happen in normal development.

---

## [Author Response]

Reviewer 1:1) Figure 4H-I. Mosaic knockdown of pk by RNAi is used to distinguish which side of the cell junction Pk and Sple isoforms reside. The authors conclude that both isoforms are proximal. This is an important result as it is different from previously published studies overexpressing Sple and should be shown more clearly. It is very difficult to distinguish the boundaries between expressing and non-expressing cells, especially considering that margin cells may contribute to the signal.

We recognize that these images require some effort to interpret. To assist readers, we have drawn outlines just inside the cell boundaries of the relevant cells in the initial panels. We prefer to not also place those outlines in the second and third panels so as to not obscure the data. We think that a motivated reader will be able to make their own interpretation and that it will agree with our description.

2) Figure 4—figure supplement 1A-B. The loading control levels are highly variable making it difficult to interpret how Pk and Sple isoform levels change over time. Perhaps tubulin is not an appropriate loading control if it levels too change over time. Can the authors plot Pk and Sple levels relative to Tub loading control? Or use V5 fluorescent intensities from confocal images to compare levels?

We have provided the requested quantification of Pk and Sple relative to γ-tubulin and presented the results as a bar chart.

3) Figure 4. The authors conclude from the localization of Pk and Sple isoforms "that distal localization of Pk^sple^ in Pk^pk^ mutants drives counter-clockwise rotation of shaft-socket pairs." In addition, the title of this section is "Localization of Pk^pk^ and/or Pk^sple^ determines handedness of helical rotation". I think these are overstated conclusions. The localization certainly correlates with the direction of rotation, but this seems insufficient to conclude this drives rotation, especially since elimination of both isoforms has very little effect on rotation.

We agree, and the wording has been changed to reflect correlation. In the conclusion, we then “suggest” that because Pk^sple^ determines counter-clockwise rotation, its localization is likely the determinant of that rotation. As discussed more extensively below, we have made revisions to the text to more clearly emphasize that Pk^pk^ does not actively determine the direction of rotation, but instead allows the core module to interpret the direction of rotation.

4) Figure 7D. This experiment shows that reversed Sple localization in Pk^pk^ mutants depends on Ds, and "we interpret this to indicate that the Ds global signal recruits Pk^sple^ to sites of enriched Ds (distal) in the absence of Pk^pk^". What about in the proximal part of the wing where bristle polarity is normal? Is Sple localized proximally, and independent of Ds? How do the authors interpret the distally restricted phenotype of Pk^pk^ mutants? Is the Ds signal stronger distally?

The localization of Pk^sple^ is discussed in the manuscript (paragraph four subsection “Localization of Pk^pk^ and/or Pk^sple^ correlates with handedness of helical rotation”; Figure 4—figure supplement 2 C-C’’’; and in the Discussion, paragraph four of subsection “Spatiotemporal dynamics and selection of Pk^pk^ versus Pk^sple^ 514 for polarity determination”). Our analyses do not have sufficient resolution to fully understand the differences between the distal wing where polarity is reversed and the proximal wing, where it is not, but we refer the reviewer to these passages for some possible explanations.

5) Coupling between the Fz-Vang and Fat-Ds pathways by Pk isoforms. "we interpret this to indicate that the Ds global signal recruits Pk^sple^ to sites of enriched Ds (distal) in the absence of Pk^pk^ (Figure 7B; compare with 7D), which drives reversal of shaft-socket orientation and therefore reversal of bristle polarity." Are Fz and Vang localizations flipped in socket cells with reversed polarity in Pk^pk^ mutants? Although this has been shown previously for wing hairs overexpressing Sple, I feel it is important to show here under endogenous Sple levels to definitely show the core PCP module is flipped, not just a downstream component.

We have now done this analysis for Vang, and indeed the core module is flipped. The data are provided in Figure 5—figure supplement 1.

Reviewer 2:1) A "helical structure of defined chirality" may not be appropriate. It seems to be an anisotropic cell rearrangement directed by distal-proximal and dorsal-ventral polarity. Such rearrangement should change the interface between shaft and socket as shown in Figure 3B and C.

Here, we presume that in saying “anisotropic cell rearrangement,” the reviewer refers to junctional rearrangement. If instead they intend it more generically then it is just a vaguer statement of generating a structure of defined chirality. With that in mind, we think that both generating a structure of helical chirality and anisotropic cell rearrangements are both valid observational descriptions of what we see, each conveying different aspects of the event. Adding that these are “directed by distal-proximal and dorsal-ventral polarity” is an interpretation. Regarding the description, we prefer emphasizing the structure, as that is better documented that the junctional rearrangement, for which we lack live imaging that would be needed to solidify that inference.

2) It is difficult to evaluate the phenotypic analyses without statistical information. What are the percentages of wings showing the representative phenotypes? How may socket cells were scored from how many wings? How many clones were analyzed and what was the percentages showing the described phenotypes? Some of them can be found in source data files. However, it should be also stated in the text.

We have provided additional statistical information in the legends for main Figure 5, 6, and 7, in the legends for Figure 2, 3, 6, and 7 figure supplements, and in several additional source data files. In addition, we have added several comments about consistency of qualitative results where appropriate. In some cases, we have not addressed “What are the percentages of wings showing the representative phenotypes?” In all cases the phenotypes we show are very consistent unless otherwise indicated (ie dsRNAi). In the Materials and methods section, we have added the sentence “For qualitative results such as expression patterns, a minimum of 20 biological replicates from at least two independent experiments were examined and representative images are shown.” We would hope that it is not necessary to state that the qualitative results we describe are consistent and that we have chosen representative images.

3) Figure 4H-I', Figure 2—figure supplement 1B, and Figure 4—figure supplement 2 were unclear. These data should be shown more clearly.

To aid in interpreting Figure 4H-I’, we have added some outlines of relevant cells that we hope will be helpful. For Figure 2—figure supplement 1B, we have added some additional colored dots and explanation. We reiterate that the junctional rearrangements are presumed, since we do not have live imaging. If the reviewer is concerned that the entirety of Figure 4—figure supplement 2 is unclear, we’re at a loss as to how to respond. In every instance we have attempted to select representative images that show as clearly as possible the feature of interest.

Reviewer 3:1) As a general comment, it might be helpful to readers to explain that in processes such as left-right asymmetry determination (analogous to the chiral morphogenesis being discussed here) there is symmetry breaking on three axes: apical-basal, and two axes in the plane of the tissue. Hence, I presume PCP specifies clockwise or anticlockwise morphogenesis by specifying polarity in one axis of the plane, in concert with a second polarity cue in the orthogonal axis in the plane (which I take to be an AP oriented cue, ultimately set up by the Hedgehog pathway?). Without this second cue, the PCP cue alone wouldn't be sufficient to specify chirality?

We agree, and a statement articulating this idea has been added.

2) Introduction paragraph three: I think consistent nomenclature for proteins in flies is capitalized Roman so should be "Prickle" and "Spiny-legs" (and probably "Pk^Pk^" and "Pk^Sple^")? (Also some genotypes change to all caps italics on figure panels e.g. compare Figure 5E-F to 5G-H).

Since these are isoforms of one protein, we think the correct designation is “Prickle^prickle^ (Pk^pk^) and Prickle^spiny-legs^ (Pk^sple^)”.

3) Results paragraph one: typo "Supressor"

Thank you.

4) Figure 4—figure supplement 1A is mislabeled? The blue asterisks ("unidentified") and the purple arrow ("Pk") should be swapped? This would then be consistent with panel B?

The arrows are correct, although the confusion may have arisen from incorrect designation of molecular weight markers (now corrected) in panel B. The unidentified band co-migrates with V5::Pk and occurs in variable intensities.

5) Subsection “Pk^pk^ and Pk^sple^ differentially interpret directional signals from Ds” imply an "indirect" mechanism for Pk^Pk^ coupling to Ft-Ds, but the Introduction (I think more in line with previous work, including from the authors) suggest an indirect mechanism for core pathway coupling to Ft-Ds via Fz/Dsh/Fmi transport with (presumably) Pk^Pk^ being required for magnitude of core pathway activity but *not* for orientation? This is a complex area, but I think it would be easier for readers if the authors stuck to one view. At least based on the data so far, we don't see any evidence that Pk^Pk^ couples, only that it enhances core pathway activity?

Both sections are intended to refer to the same model, in which Ft/Ds couples to the core pathway indirectly via microtubule directed transport when Pk^pk^ is participating. We have adjusted the language to better capture this idea throughout the manuscript so as not to imply more than what we mean to say. Specifically, we do not wish to suggest an active role for Pk^pk^ in coupling, but rather the microtubule mechanism couples when Pk^pk^ is used for core signaling (and evidently to some extent in the absence of either Pk isoform).

6) Paragraph three of subsection “Pk^pk^ and Pk^sple^ differentially interpret directional signals from Ds”: the authors deduce that in the absence of Pk^Pk^, Pk^Sple^ is recruited to distal cell edges by distally localised Ds, but I find this confusing (at least without further clarification) as Merkel et al. seemed to show pretty clearly that – at least in the wing blade – Ds is oriented primarily on the AP axis from 16h APF onwards, and trichomes orient primarily on the AP axis in pk^pk^ as well (e.g. Merkel et al. and Gubb et al.). So I would expect Sple to be recruited largely to AP cell edges (maybe with a slight distal bias?) not to distal cell edges. But maybe something happens on the edge of the wing blade to tilt the Ds polarity axis more PD? What do the authors think?

The Merkel analysis of Ds orientation didn’t extend to the wing margin, so it is not directly pertinent. We furthermore suggest that the Merkel et al. analysis doesn’t have sufficient resolution or precision to make this assessment. And finally, the reviewer is correct that only a subtle PD bias would be sufficient. Of note, in the region where bristles are reversed in *pk^pk^* mutants, the hairs nearest to the margin are similarly reversed, rather than being primarily AP as the reviewer suggests, and indeed notes themself in the following point 7. Please see Author response image 1.

7) Paragraph four subsection “Pk^pk^ and Pk^sple^ differentially interpret directional signals from Ds”: this appears to be a key sentence, as here the authors appear to argue that ds knockdown affects a Pk^Pk^-mediated signal for bristle polarity, however I completely failed to follow the logic that indicates Pk^Pk^ is providing a polarity cue rather than contributing to polarity magnitude (see paragraph two of subsection “Pk^pk^ versus Pk^sple^ isoform expression determines the direction of rotation” and Table 3) and suppressing effects of "ectopic" Sple.– The key piece of data is (I think) that ds- gives reversals of bristle polarity, and (perhaps surprisingly) these are suppressed by loss of Pk and loss of Pk and Sple, but not by loss of Sple alone. I may be missing something important here, but these results seem to say that having Pk activity causes a disruption in polarity in the absence of Ds, but does nothing to support the view that Pk orients polarity?

The reviewer is overinterpreting our statement in the cited lines. Here, we’re only trying to explain the random local domains of P and D polarity in *ds* knockdown wings. These depend on Pk^pk^, though we cannot see the Pk^pk^ localization for technical reasons. We’re not saying Pk^pk^ orients polarity; we’re simply arguing that it allows the randomly oriented, locally correlated domains to form. Language clarifying this claim has been added to the manuscript.

– Having given it some thought, I think the key to understanding these bristle phenotypes is the original observation of Gubb and Garcia-Bellido, 1982 that "Trichomes and chaetae express the same polarity" i.e. the bristles do what the adjacent trichomes do. In ds we know trichomes reverse in the mid-anterior wing margin (Adler et al., 1998) and this seems to be due to altered core PCP in turn produced by disrupted cell flows, oriented cell divisions etc (Aigouy et al., Merkel et al.) [other explanations for the swirling phenotype are possible but don't alter the argument I think]. In ds double mutants with the core pathway, trichomes show a fz/in pattern (Adler et al., 1998) and point broadly distal on the anterior wing margin, and we see bristles doing the same in ds pk^pk-sple^ i.e. the phenotype of core loss-of-function is epistatic to that of ft-ds for trichome polarity. In ds pk^pk^ we also see trichomes pointing distally as do the bristles (e.g. Hogan et al., 2011), presumably because Sple isn't being mislocalized by Ds (as also suggested in the text by the authors here) and we're seeing either a wild-type or fz/in loss-of-function polarity phenotype?

We agree that bristles and hairs are under the same control at the AWM, and that polarity is transmitted between them, as discussed above. But just saying that bristles and hairs share the same polarity is hardly an explanation for the direction we see in various mutants, since it doesn’t explain why either the hairs or bristles behave as noted. As an aside, we’re unconvinced that cell flows are the answer, but that is another discussion. The essence of our finding is that in this region, Pk^sple^ expression comes up earlier than the rest of the wing, accounting for the cleaner reversal when Pk^pk^ is removed.

Comments related to trichome polarity:1) The authors also revisit the issue of whether Ft-Ds instruct trichome polarity in Figure 7G-K. However, these results just seem to agree with the view that the core loss-of-function trichome polarity phenotypes are epistatic to loss- or gain-of-function ft-ds trichome polarity phenotypes. There is no dispute that dll>2x-ds makes trichome polarity swirl (and as they normally point distally, any swirling can be interpreted as "reversal"), but I think it is hard to prove or disprove whether this is direct effect on the core pathway or an indirect effect on the complex process of wing morphogenesis (c.f. work from Eaton lab) that alters core polarity. I don't think it's a surprise that loss of fz or pk-^sple^ suppresses this phenotype (epistasis noted by Adler et al., 1998) or that pk^sple^ does not (this leaves core pathway intact).

We agree entirely that our data are consistent with epistasis between Ft-Ds and core PCP signaling. Epistasis, done properly, implies an upstream-downstream relationship which we indeed believe to be the case between these two systems when pk^pk^ is participating in core function.

However, we wish to go one step further, and address whether Ft-Ds is *instructive* during Pk^pk^ function, and loss-of-function epistasis is not sufficient to make this assessment. Demonstrating an instructive role requires reorganizing the Ft-Ds signal in a defined way and asking if the core response follows the predicted input. We therefore identified a condition, *dll>2x-ds*, in which a reorganized Ds gradient redirects polarity. The reviewer correctly notes that there are some swirls in this pattern, and that by definition if there are some swirls there must be areas if reversed polarity. But one way to create swirls is to induce areas of reversed polarity, and we argue that *dll>2x-ds* does exactly that. To try to make the point more convincingly, we now provide an image showing a much larger portion of such a wing, and highlight the region where the reversed gradient is most pronounced (new Figure 7—figure supplement 1E). Note that in the majority of this region, hairs are reoriented to point away from the margin, as expected based on the shape of the *dll* expression domain, whereas swirls appear primarily at the borders of this domain where the reversed polarity confronts regions that are not reversed because ds is either expressed at too low a level or too high a level to produce a meaningful gradient. In addition, we’ve moved the location of the close-up images in Figure 7G-K away from the L3-L4 intervein space where we sometimes saw distortions of the reversal, as in this example, perhaps due to the veins, or possibly due to the geometry of the *dll* gradient, to the L4-L5 intervein space where the effect is more cleanly seen.

The reviewer also focuses on “whether this is direct effect… or an indirect effect…” We hope the reviewer will agree that the questions of epistasis and of instructive function imply nothing at all about how direct or indirect the signal is between the two systems; how direct or indirect is instead a quite distinct question. If, as the reviewer suggests, Ft-Ds acts on core PCP by modulating wing morphogenesis (c.f. work from Eaton lab), this would simply provide an explanation for the epistasis and instructive function. In this manuscript, we make no specific argument about the mechanism – we simply argue that the Ft-Ds system instructs directionality of the core system when Pk^pk^ is participating. However, for this reviewer, we note that we do not favor an Eaton mechanism, such as perturbation of cell flow and shear, because these physical processes would be expected to alter the mutant polarity patterns as well as the wildtype pattern, yet this is not observed. The fz^fz^, *pk^pk-sple^* and *pk^pk^* patterns are essentially unperturbed by *dll>2x-ds.*

The interesting question is why the dll>2x-ds; pk^pk^ phenotype looks like the pk^pk^ phenotype. The authors describe this as a "suppression", and seem to conclude that this means Pk^Pk^ couples to Ft-Ds, but I don't follow the leap of logic. I think the accepted view is that in pk^pk^ the residual population of Pk^Sple^ is oriented by endogenous Ft-Ds generating the familiar pk^pk^ AP oriented trichome pattern, and it appears that this AP orientation persists in a dll>2x-ds background. This persistence is interesting, but seems to be more about how and when Pk^Sple^ couples to Ft-Ds, and doesn't seem to reveal anything about whether Pk^Pk^ can couple to Ft-Ds. Have I missed something?

As discussed above, for the purposes of our argument, we suggest the important observation is that *dll>2x-ds does* redirect the polarity pattern when Pk^pk^ is present – if the systems are uncoupled, as has been argued by others, this should not occur.

There is, however, an interesting point to be made regarding the question the reviewer brings up: why the *dll>2x-ds; pk^pk^* phenotype looks like the pk^pk^ phenotype. Indeed, it has been argued that in the absence of Pk^pk^, Pk^sple^ contributes to the polarity pattern, and this can be seen in the difference between the polarity patterns of *pk^pk^* and *pk^pk-sple^* mutants. We see relatively subtle differences between the *pk^pk^* pattern and the *dll>2x-ds; pk^pk^* pattern, consistent with our data that there is little Pk^sple^ expression when hair polarity is established. One exception is near the anterior wing margin, where Pk^sple^ expression does become significant at this time. In the *pk^pk^* pattern, hairs (and some bristles) point proximally, and in the *dll>2x-ds; pk^pk^* pattern, they mostly point distally (as would be predicted with reversed Ds gradient and Pk^sple^ control of core response).

**Author response image 1. respfig1:** *pk^pk^*and dll>ds;*pk^pk^*anterior wing margins.

2) Overall I feel it's a pretty shaky case that Ft-Ds is directly orienting the core in these experiments or that Pk is acting to couple the core to Ft-Ds. Of course if you did want to argue that Ft-Ds orient the core in the relevant region of the wing blade to affect either distal trichome polarity or the triple row bristles, I think you would have to carefully reconcile the following published observations:i) Ft-Ds polarity seems to be AP while core is PD, so they don't point the same way? (A timing issue maybe?)ii) flattening Ds and Fj gradients seems to give normal polarity? (Also of bristles?)iii) ft dachs and ft UAS-ft-intra and ft UAS-wts give normal trichome polarity? (Triple row bristles also point distally in these genotypes?)However, this all seems like a different argument, not pertinent to helical morphogenesis.

Again, we reiterate that we are not claiming that Ft-Ds is “directly orienting the core.” We argue that Ft-Ds orients the core system while under Pk^pk^ control but make no statement about how direct or indirect that signal is. Nor do we argue that “Pk is acting to couple the core to Ft-Ds.” Indeed, we have previously proposed a rather indirect model in which Ft-Ds orient polarized microtubules on which vesicles containing core proteins traffic. In this model, coupling is indirect and is not mediated by Pk^pk^.

Nonetheless, we will address the points raised by the reviewer.

(i) Orientation of the Ft-Ds system has not been examined at the anterior wing margin. Published data only measure to a distance of several cells away from the margin. Still, looking at the Eaton data (Merkel et al.), both the Ft-Ds and core systems evolve from a radial pattern, but the core system becomes parallel while the Ft-Ds system does not. As we suggested originally in Ma et al., 2003, the tendency of the core system toward local alignment is stronger than the coupling to the Ft-Ds system. Thus, the early Ft-Ds orientation and perhaps a subtle remaining P-D orientation at later times could be sufficient to produce the effects we see.

(ii-iii) While these genotype-phenotype relationships are interesting, we believe they address a different question, specifically, whether there is some other information that is revealed when Ft-Ds is partially or wholly disabled. There is little doubt that perturbing the Ft-Ds signal in a variety of ways and in several tissues impinges on core PCP directionality. The listed genotypes attempt to completely remove this information. If that is successful (we’d argue that’s not entirely clear), what remains is indicative of what other information might orient core PCP, but does not bear on how Ft-Ds, when active, interacts with the core PCP system. Yes, we’d like to better understand these observations, but that is not necessary to interpret the experiments in this manuscript.

3) Nevertheless, in light of the above, I worry about the statement in the Abstract "Pk^pk^ and Pk^sple^, determines right- or left-handed bristle morphogenesis, each by coupling Frizzled/Vang signaling to the Fat/Dachsous PCP directional signal in opposite directions". On one hand I don't follow the argument showing Pk^Pk^ is coupling to Ft-Ds or that Ft-Ds is determining bristle polarity. The fact that bristle polarity is disrupted in ds mutants doesn't necessarily imply Ds is instructive. I agree that in the absence of Pk^Pk^, Pk^Sple^ is probably reversing bristle polarity by coupling to Ds – but this doesn't happen in normal development.

We agree the wording in the Abstract was problematic, although for a somewhat different reason, and it has been changed. We do not aim to imply an active role for Pk^pk^ in coupling, but rather that Ft-Ds couples to the core PCP system when the core system utilizes Pk^pk^. It is also correct that “The fact that bristle polarity is disrupted in ds mutants doesn't necessarily imply Ds is instructive.” As discussed above, the reorienting response of the core PCP system, and of cellular polarity, upon introduction of an alternately produced Ds gradient is the basis for claiming Ft-Ds is instructive. We hope that this discussion, and the improved language in the revised manuscript, is persuasive.